evolution, behaviour

beak shape, foraging, thermoregulation, bird song, trade-off, Meliphagidae

**Author for correspondence:**
Nicholas R. Friedman
e-mail: nicholas.friedman@oist.jp

†These authors contributed equally to this study.

# Evolution of a multifunctional trait: shared effects of foraging ecology and thermoregulation on beak morphology, with consequences for song evolution

Nicholas R. Friedman[1,4], Eliot T. Miller[2], Jason R. Ball[1], Haruka Kasuga[1,3], Vladimír Remeš[4,5,†] and Evan P. Economo[1,†]

[1]Biodiversity and Biocomplexity Unit, Okinawa Institute of Science and Technology Graduate University, Onna-son, Okinawa, Japan
[2]Cornell Laboratory of Ornithology, Cornell University, Ithaca, NY, USA
[3]Graduate School of Information Science and Technology, Hokkaido University, Sapporo, Hokkaido, Japan
[4]Department of Zoology and Laboratory of Ornithology, Faculty of Science, Palacký University, Olomouc, Czech Republic
[5]Department of Ecology, Faculty of Science, Charles University, Prague, Czech Republic

NRF, 0000-0002-0533-6801; JRB, 0000-0003-2311-1355

While morphological traits are often associated with multiple functions, it remains unclear how evolution balances the selective effects of different functions. Birds' beaks function not only in foraging but also in thermoregulating and singing, among other behaviours. Studies of beak evolution abound, however, most focus on a single function. Hence, we quantified relative contributions of different functions over an evolutionary timescale. We measured beak shape using geometric morphometrics and compared this trait with foraging behaviour, climatic variables and song characteristics in a phylogenetic comparative study of an Australasian radiation of songbirds (Meliphagidae). We found that both climate and foraging behaviour were significantly correlated with the beak shape and size. However, foraging ecology had a greater effect on shape, and climate had a nearly equal effect on size. We also found that evolutionary changes in beak morphology had significant consequences for vocal performance: species with elongate-shaped beaks sang at higher frequencies, while species with large beaks sang at a slower pace. The evolution of the avian beak exemplifies how morphological traits can be an evolutionary compromise among functions, and suggests that specialization along any functional axis may increase ecological divergence or reproductive isolation along others.

## 1. Introduction

The extent to which traits are optimized for different functions is a core question in evolutionary biology. Many studies of adaptation have focused on a single proposed function and measured its effect on trait evolution [1]. However, morphological structures in nature are often associated with multiple functions or behaviours [2]. It remains unclear how selection to optimize trait values for each function should affect the overall structure and its components over evolutionary timescales, as selection fluctuates and varies across space during diversification [3]. One possibility is that different components of a trait should correlate with separate functions such that each selective optimum is not mutually exclusive with others. Alternatively, constrained or inherently mutually exclusive optima may produce traits that are a compromise between functions [4,5]. Studies are needed that can compare the relative contributions of different functions over evolutionary timescales to reveal how the effects of these functions on morphological evolution are balanced.

Bird beaks are an excellent example of a multifunctional trait. Birds use their beaks for foraging, and selection on beak morphology related to this function is a textbook example of evolution [6,7]. However, birds also use their beak for nest building [8], preening and parasite removal [9], singing [10] and thermoregulating [11,12]. Both population-level [13–15] and comparative studies [16–18] have identified correlations between beak phenotypes and the functions described earlier. While previous tests of these relationships have provided important insights into the evolution of beak morphology, they have mostly tested a single hypothesized function (see [19,20]). Integrating these different functions in a single-study system to identify how beaks are related to their many functions is thus a means to improve our understanding of how multifunctional traits should evolve.

That morphological structures in general, and beaks in particular, often evolve in an integrated fashion imply that evolutionary trade-offs should exist and that it should be difficult to arrive at each function's univariate optimum simultaneously [21,22]. However, it is worth considering that species' traits may be capable of evolving such that their functions are *split* between elements of beak shape. For example, it is conceivable that selection on foraging behaviour might drive the evolution of relative beak length or depth (as in [6,19]), but that selection on thermoregulation ability might independently drive the evolution of beak size (as in [23]). Alternatively, functions may be *shared* across elements of beak shape for two reasons. First, this might be expected if morphological integration of the beak may be so strong that it constrains the availability of phenotypes that can satisfy specialization for multiple functions [1,24,25]. Indeed, Bright *et al.* [22,26] found that integration between the beak and skull explained more variation in beak shape than diet. If morphology of the beak alone is similarly integrated, it would evolve more like a single integrated trait and a compromise among its many functions. Second, even without developmental constraints, different functions can select for optima that are mutually exclusive such that a trade-off between the functions must arise (i.e. a jack of all trades is a master of none [27–29]).

Aside from functions relating to ecological niche, beaks often also function in visual and acoustic signalling. In songbirds especially, beak size and shape are related to vocal performance [10]. Several recent studies in woodcreepers (Furnariidae) have found that song characteristics are influenced by morphology at broad taxonomic scales [16], even when accounting for the direct influence of the environment on signal evolution [30]. Beak morphology may thus be a means by which niche evolution indirectly influences divergence in mating signals [31]. To better clarify the role that niche evolution plays in divergence, studies are needed that can connect changes in function to changes in morphology, and changes in morphology to changes in animal signals. Here, we use a comparative study of honeyeaters (Meliphagidae) to test this framework.

Honeyeaters are a diverse radiation of songbirds (Passeriformes) that belong to a lineage sister to core oscine lineages, and they have remained almost completely confined to Australasia [32]. Having originated in a wet subtropical environment [33], they now inhabit broad climatic and biotic gradients in Australasia, from inland deserts to temperate sclerophyll forests and tropical island rainforests. The honeyeaters have historically been confined to a single zoogeographical realm to the southeast of Wallace's Line [34,35], making them a convenient replicate of songbird evolution for studying change within a single biogeographical arena [36,37]. They vary widely in their diet, size and climate preferences and aspects of their beak morphology have previously been associated with their diet [38] and with winter temperatures (i.e. Allen's rule [17,39,40]). In adapting to Australia's arid interior, they have evolved divergent morphologies and behaviours [41].

We used a geometric morphometric approach to quantify the beak shape and size in 101 species of honeyeaters. We integrated this with foraging behaviour from [41], song characteristics (frequency and pace) and climate to examine the multivariate nature of beak shape and size optimization. First, we examined whether there is a detectable relationship between each function and beak morphology in honeyeaters when controlling for other predictors. Second, we quantified the extent to which axes of beak morphology are either shared or split between different functions. Finally, we tested the effect of niche evolution on vocal traits as mediated by beak size and shape.

## 2. Material and methods

### (a) Geometric morphometrics

We measured 525 specimens from the Natural History Museum in Tring, UK, to describe variation in beak shape among 101 honeyeater species (see the electronic supplementary material, appendix S1; mean 4.8 specimens per species). We photographed each specimen's beak under standardized focal distance and lighting conditions (Nikon D80 camera with Nikon 105 mm Micro lens). Specimens were aligned relative to the camera using an adjustable stage such that their midsagittal plane was in line with the camera's focal plane, and a measurement standard was included in each photo. We chose a set of four landmarks and 20 semi-landmarks that defined the outline of the rostrum and nare (figure 1). All landmark measurements were performed by N.R.F. We did not include information on the outline of the mandible or the portion of the rostrum posterior to the nares, as these features were often obscured by the rostrum and head feathers. Future studies may address this issue using either disarticulated skeletal material or penetrative scanning methods (e.g. computed tomography).

We aligned these landmarks and semi-landmarks using a generalized Procrustes analysis performed with the *geomorph* package in *R* (v. 3.0.6; [42]). Procrustes distances were used as the optimization criterion for semi-landmark alignment. Outliers beyond the upper quartile of Procrustes distances from the mean were re-measured to confirm they reflected real variation in shape and not digitization errors. Following Procrustes alignment, we produced species averages using the *mshape* function in *geomorph*. We extracted principal components (PCs) of beak shape variation for use in subsequent analyses and visualized their loadings using thin-plate spline warp grids (figure 1). We estimated beak size from photographs by taking the centroid size of the landmarks following Procrustes analysis, while using a landmark measurement of the standard included in the photo. This method is a common way to measure the shape-free estimate of size and is commonly employed in geometric morphometrics [43].

### (b) Foraging behaviour

We used data collected as described in [41] to describe foraging behaviour of honeyeater species. In brief, these data were collected from 9595 field observations of foraging behaviour and

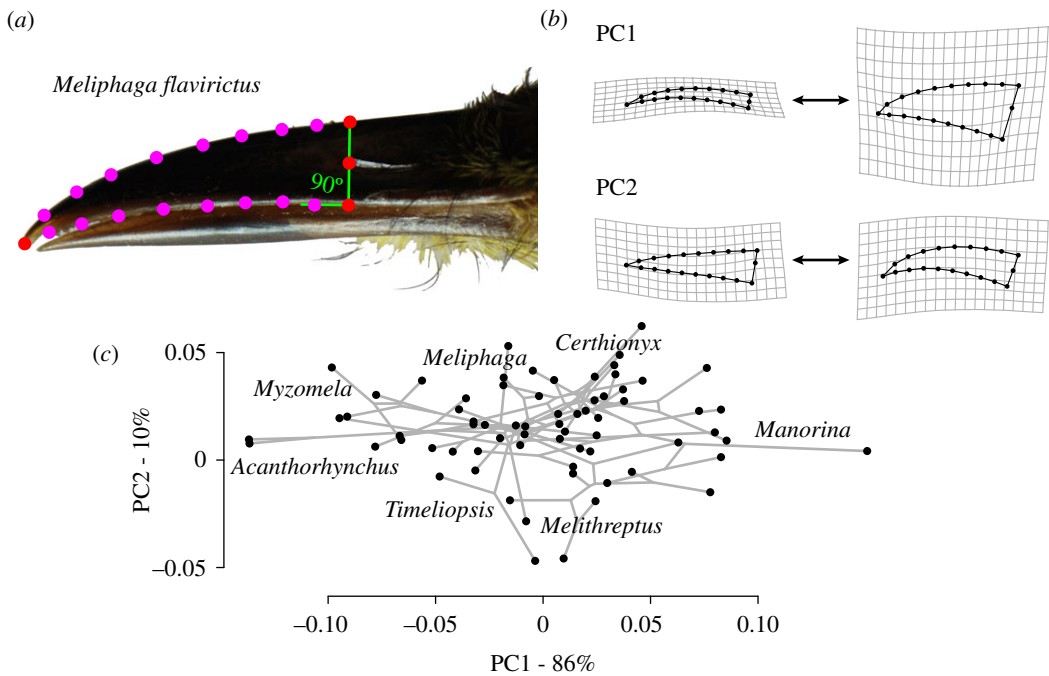

**Figure 1.** (a) Yellow-gaped honeyeater specimen illustrating positions of landmarks (red) and semi-landmarks (magenta). Semi-landmarks were spaced at equal intervals between landmarks using *TPSdig*. (b) Relative warp grids showing the extreme values of the first two principal component axes (PC1 and PC2). PC1 is referred to as 'depth', and PC2 is referred to as 'curvature' throughout. (c) Phylomorphospace of honeyeater beaks, using the first two PC axes, which together account for 95% of shape variation. Genera with divergent phenotypes are noted.

dietary preferences for 74 species of honeyeaters in every territory of Australia. We ordinated variables from these behavioural observations using a phylogenetically corrected principal component analysis in *phytools* (pPCA; [44]). The results of this ordination are as described by Miller *et al.* [41]. The first pPC axis described a continuum from species that glean insects from leaves to species that rely on nectarivory and occasional aerial attacks on flying insects. The other pPC axes described the extent to which species foraged in the canopy versus on the ground (as in *Epthianura*) and the extent to which they employed comparatively rare, specialized behaviours such as foraging on insect cases and using the beak to lever open rolled substrates (gaping; see the electronic supplementary material, figure S1 for pPC biplots).

## (c) Song analysis
We measured 711 recordings available in public databases at the Macaulay Library at the Cornell Laboratory of Ornithology, the Naturalis Biodiversity Center and the Australian National Wildlife Collection (see the electronic supplementary material, appendix S2). Our song analysis protocol closely followed those described in previous studies of song evolution [45,46]. We treated each recording as a separate singing bout and measured no more than one song type per individual to maximize the independence of our samples. We excluded recordings that were deemed of poor quality, particularly those in which sounds in the background prevented the accurate measurement or identification of the target species (identifications were made using recordists' notes and descriptions in the Handbook of the Birds of the World [47]). For the purposes of this study, our operational definition of a 'song' was any vocalization that included tonal elements, was longer than 1 s in duration and was preceded and followed by intervals greater than 1 s [45]. We included intervals greater than 1 s in songs only if they were part of a consistently repeated pattern of note types (e.g. A … BC).

We generated spectrograms in RAVEN PRO v. 1.5 (Cornell Laboratory of Ornithology 2014) using a window size of 256 samples. On each spectrogram, we recorded the start and end time of the song and each note, as well as the maximum and minimum frequency of each note (for at most 30 notes). From these data, we calculated two metrics of song frequency: mean note minimum frequency and mean note maximum frequency. We also calculated an estimate of 'song pace' following [46], which was the total number of notes in each song divided by the song's duration. These metrics were chosen among the many that have been previously described [45,46] based on previous studies showing that they should be most influenced by changes in beak morphology, as well as their general applicability across a broad taxonomic range [10,48–50].

## (d) Climate
To describe the thermoregulatory challenges faced by species in different habitats, we estimated the average winter minimum temperature and summer maximum temperature for each species based on their breeding range. We used rasterized climate data from the Bioclim dataset (1960–1990; bio5, bio6 [51]), averaging across cells included in each species' range using the *R* package *raster* (v. 2.6-7 [52]). Range maps were provided by BirdLife International and NatureServe [53]. Species average summer maximum temperatures ranged from 20 to 38°C, and winter minimum temperatures ranged from 3 to 22°C.

## (e) Phylogeny
Marki *et al.* [54] inferred a time-calibrated phylogeny for the infraorder Meliphagides using BEAST [55]. Their study was based on a supermatrix assembled from four nuclear loci (two introns and two exons) and five mitochondrial loci. We used this phylogeny, pruned to include only species in our focal group, for all analyses described below.

## (f) Comparative methods
We performed regression tests to address (i) the relationship between foraging, climate and morphology and (ii) the relationship between morphology and song. In the former set of tests,

morphology is the response variable, and in the latter set, song is the response variable. We corrected for similarity owing to shared evolutionary history and included an effect of body size [56] on beak size (either as covariate or body size residuals when that was not possible) in each analysis.

We employed four sets of comparative analyses: (i) phylogenetic generalized least-squares regressions (PGLS [57]) were used to test the relationship between a set of predictor variables and a single response variable (a feature of either beak morphology or song behaviour). We performed these tests using an estimated lambda parameter to control for the amount of phylogenetic effect in the model residuals [58]. We scaled variables by their standard deviations and centred them on zero to produce standardized regression coefficients. Because song behaviour is often variable in passerine species, we repeated our PGLS analyses involving song traits using a method that accounted for this intraspecific variation [59].

However, relationships among climate, behaviour and morphology may be complex [41,60]. Hence, (ii) we used a phylogenetically corrected path analysis to disentangle these relationships and their effects on beak shape. This was accomplished using the R package *phylopath* (v. 1.0.1 [61]). We tested between path models varying on two axes: first, function–trait relationships were organized such that functions were either split or shared between different axes of trait variation (figure 2*a* versus *b*); second, relationships between functions (e.g. between temperature and foraging ecology) either were or were not included (figure 2*a* versus *c*). We controlled for the effect of body size on beak size in the path analysis by including the beak size residuals of their allometric regression. We evaluated these models by comparing their *C*-statistic information criterion (CICc), which describe model fit while taking into account the number of parameters/paths [62].

Because shape is a highly dimensional trait, we also employed several multivariate approaches. We performed (iii) a phylogenetic Procrustes analysis of variance (ANOVA) that treated Procrustes-aligned beak shape as a response variable [63]. In this analysis, we included all PCs of foraging behaviour, as well as both summer and winter temperatures, body size and beak size as predictors. We performed this analysis using a Brownian motion model in the *geomorph* function *procD.pgls* and 5000 iterations of resampling for significance testing. Effect sizes from this analysis are intended to describe the overall contribution of each predictor variable in explaining beak shape evolution in honeyeaters. Pseudo-$R^2$ values were calculated for the beak size as a comparison with beak shape, based on a multivariate phylogenetic regression implemented in *phylopath*.

We also performed (iv) a two-block partial least-squares analysis using a matrix of multivariate foraging behaviour traits as a predictor and Procrustes-aligned beak shape as a response variable. This analysis was intended to test for and measure the degree of overall covariation between these highly dimensional traits.

## 3. Results

The shape of honeyeater beaks varied among species primarily in their depth and elongation, which was the first PC axis and explained 86% of variation (figure 1; hereafter 'depth'). Along the PC1 axis, low values correspond to long and slender beaks exemplified by spinebills (*Acanthorhynchus*), whereas high values correspond to short stocky beaks exemplified by miners. Low values of PC2 correspond to straight beaks as in *Melithreptus* species, whereas high values correspond to highly curved morphologies like those seen in myzomela (10% of variation; hereafter 'curvature'). We also observed variation in the degree of tapering towards the

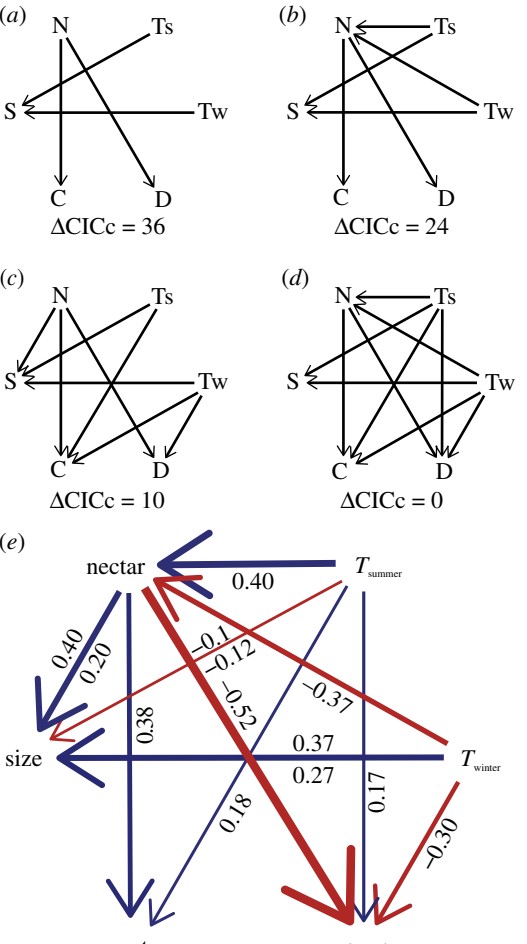

**Figure 2.** Phylogenetically corrected path analysis models. On the top versus bottom are models describing hypotheses in which axes of beak variation are split into different functions (*a,b*) or shared among functions (*c,d*). Predictors described in (*e*) are listed as abbreviations in models shown above. Here, we describe beak shape PC1 as 'depth' and PC2 as 'curvature' to reflect the positive direction of each axis. On the left and right are models describing hypotheses where functions exclude interfering indirect effects (*a,c*) or include them (*b,d*). Model fit is assessed by the *C*-statistic information criterion (CICc) following [62]. The best fitting model is shown in (*e*), with red arrows indicating negative associations and blue arrows indicating positive ones; values shown below each arrow refer to correlation coefficients; values estimated for body size residuals of beak size are shown above arrows.

distal end of the beak (PC3; 2%; hereafter 'tapering'; electronic supplementary material, figure S2). More PC axes were recovered, but none explained greater than 1% of variation.

Phylogenetic regressions revealed many relationships among beak morphology, foraging behaviour and climate. In particular, beak depth (PC1) showed significant relationships with foraging pPC1 such that species specializing on nectar had elongated beaks (PGLS: $p < 0.001$; see the electronic supplementary material, table S2; figure 3) and with winter temperatures such that the species inhabiting regions with cold winter had less elongated beaks (PGLS: $p < 0.01$). Beak curvature was similarly correlated with foraging pPCs 1 and 2 such that species foraging for nectar tended to have curved beaks (PGLS: $p < 0.05$), and species foraging on insect cases (bagworms, etc.) tended to have straight beaks (PGLS: $p < 0.001$). Species with beaks that tapered at the end tended to forage on or near the ground, whereas species

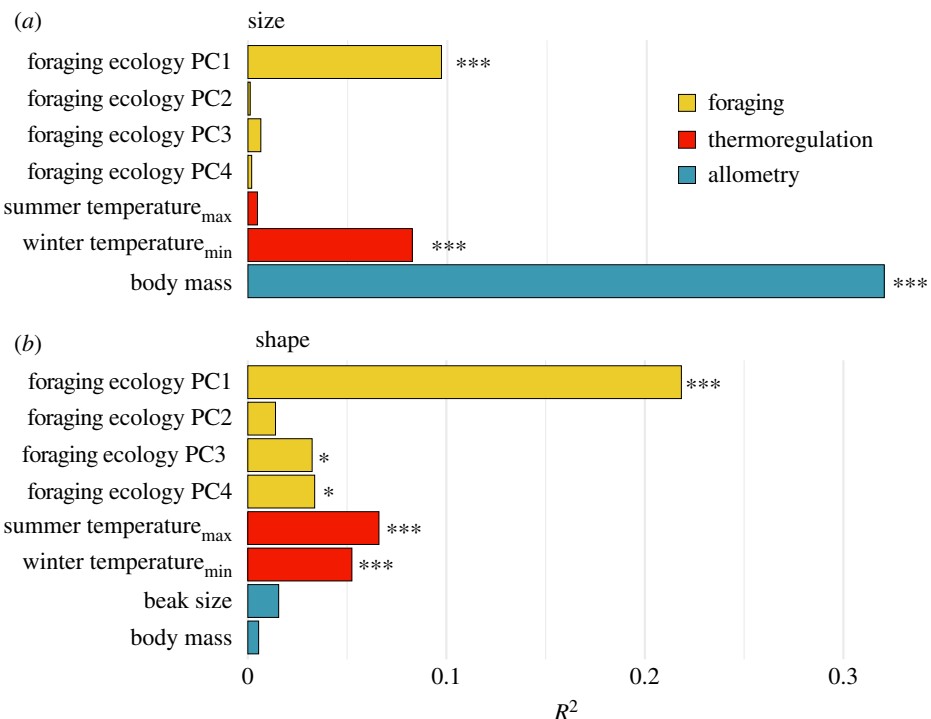

**Figure 3.** Comparison of beak size and shape variances explained by different predictor variables. (a) Values are derived from PGLS analysis of beak size. (b) Values are derived from a Procrustes PGLS analysis of beak shape that is not restricted to a single PC axis [63]. *$p < 0.05$; ***$p < 0.001$.

with beaks that tapered throughout tended to forage in the canopy (PGLS: $p < 0.01$). Because both foraging behaviour and shape are multivariate traits, we also performed a two-block partial least-squares analysis comparing their relationship, which showed a strong correlation between these two character sets ($r = 0.625$; $p = 0.003$).

To assess the extent to which the evolution of beak shape was explained by diet, climate and body size predictors, we used a phylogenetically corrected path analysis. In this analysis, models with traits linked to multiple functions were a better fit to the data (figure 2c,d). In the best fitting model, climate and foraging ecology influenced both beak size and beak shape (figure 2e). However, foraging ecology had a greater effect on shape and a slightly greater effect on size. Varying the type of body size correction used produced few qualitative changes; winter temperatures had a marginally stronger effect on beak size than foraging when size correction was removed, and both winter temperatures and foraging had marginally reduced effects on beak size when body size was included as a covariate (electronic supplementary material, figure S3). Examination of the relative variances explained for beak size and multivariate beak shape also showed that, other than allometry, winter temperatures and foraging ecology had the greatest effects on beak size, and foraging ecology had the greatest effect on the beak shape (figure 3). Both summer and winter temperatures had a significant effect on overall beak shape, but the effect of summer temperatures was greater in our Procrustes PGLS analysis (figure 3). However, when including and correcting for indirect effects (summer temperatures predicting nectarivory), winter temperatures showed a greater effect on beak morphology (electronic supplementary material, figure S4). Curvilinear relationships with temperature were in some cases a better fit to our data (electronic supplementary material, table S3 and figure S5), as in [11]. In total, these

variables explained 51% of the variance for the beak size and 43% for the beak shape.

We found significant relationships between two aspects of beak morphology, size and depth and song characteristics in honeyeaters (figure 4; electronic supplementary material, figure S6). In particular, species with larger beaks sang at a slower pace (PGLS: $p < 0.001$), and species with more elongated beaks exhibited a lower maximum frequency (PGLS: $p < 0.01$). Species with elongated beaks also sang at a slower pace when correcting for body size instead of beak size (electronic supplementary material, figure S7). Other characteristics of beak shape, curvature and tapering, showed no significant effects on song characteristics. As expected, species with larger body sizes also exhibited lower minimum frequencies.

## 4. Discussion

### (a) Relationship with foraging ecology and climate

Our results indicate that the evolution of both beak size and beak shape in honeyeaters was driven by a set of trade-offs among allometry, thermoregulation and foraging ecology. Diet and foraging ecology had a slightly greater effect on beak size than winter temperatures in both types of analyses performed. Previous studies have shown a relationship between climate and beak size [39,64] and have identified how the beak's thermoregulatory function explains this variation [15,23]. However, previous studies have also identified beak morphology as a subject of at times intense natural selection in association with foraging ecology [6,65]. Our findings suggest that selection based on both of these functions is evident over evolutionary timescales.

Beak shape was correlated with both climate and foraging ecology, but diet in particular explained the most variation in

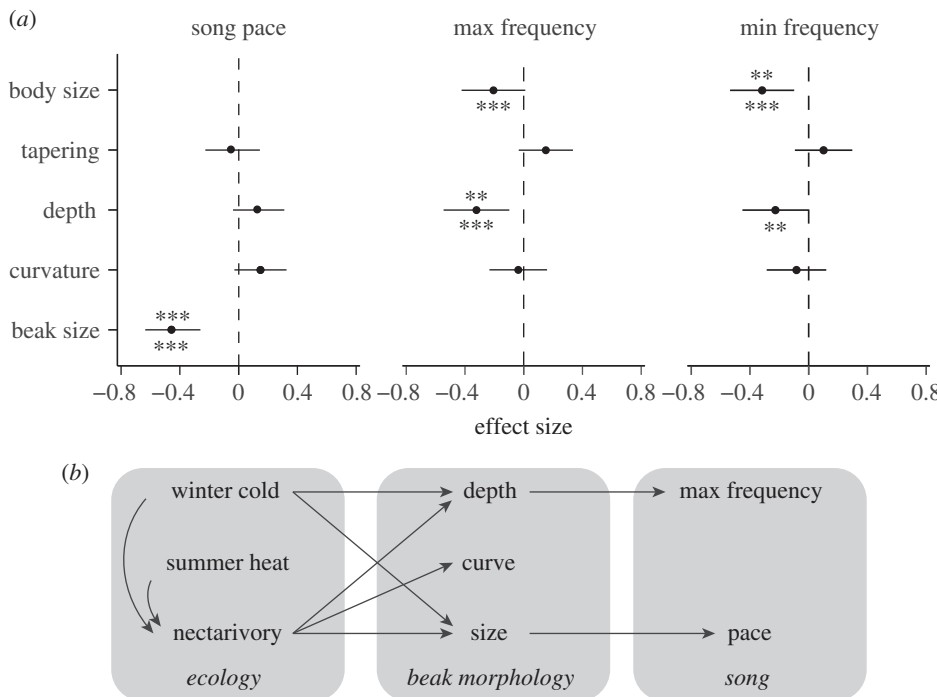

**Figure 4.** (a) Forest plot of effect sizes (standardized β) and their 95% confidence intervals for the effect of beak morphology on three different song metrics, as derived from multivariate PGLS analysis. Asterisks below bars represent significance values assessed through bivariate regressions including correction for intraspecific variation in song characteristics. (b) Diagram of general conclusions. **$p < 0.01$; ***$p < 0.001$.

this trait overall (figure 3). This result echoes that of a recent study in waterfowl, which showed that the beak shape was strongly correlated with diet [18]. However, while Olsen [18] found that diet explained 64% of variation along the first PC axis of beak shape, we found that diet explained only 23% of variation. This may suggest that major changes in foraging ecology in honeyeaters do not necessitate changes in the beak shape; the prevalence of nectarivory in most species could make changes in the morphology of their brush tongues as important as beaks for survival, if not more so [66,67]. Similarly absent from this and most other morphological analyses is consideration of musculature in the avian head and its association with foraging ecology [68]. Finally, as suggested by Miller *et al.* [41], mismatch between morphology and function may be ameliorated by changes not in form but in behaviour. Precise quantifications of trophic interactions are difficult to assemble for large taxonomic samples, which is a limitation for this and other comparative studies. While we were able to include information on flower length in our analysis, fine-scale data on the size and the identity of prey items require data from stomach contents in addition to behavioural observations [69].

Scaling with body size explained a significant amount of variation in beak size, as expected [70], but this was not the case for beak shape, whose relationship to body size is less straightforward. Previous studies of beak shape in relation to allometry have found significant relationships with far greater explanatory power than we report [22,71]. Bright *et al.* [22] reported that allometry explained more variation in beak shape than diet in raptors. The relatively weak effect of allometry on beak shape we observed in honeyeaters may thus be exceptional. While in many lineages foraging niche tends to change with the body size [72], these traits appear to be relatively decoupled in honeyeaters. Indeed, both the smallest and largest honeyeater species on the Australian continent (*Myzomela sanguinolenta* and *Anthochaera*

*carunculata*) are observed to take nectar from bottle-brush flowers, although they survive by different strategies [73].

## (b) Evolution of a multifunctional trait

We compared the goodness of fit for models including trait–function relationships that were either split (each axis of variation mapped to one function) or shared (each axis of variation mapped to multiple functions). Shared models consistently performed better (figure 2), suggesting that the evolution of beak morphology is shaped by multiple functions, even when reduced to individual elements of the beak shape. The reduction in species' traits to a single function is a classic critique of the search for adaptive optimality in organismal design [1]. Our analysis tests and supports the notion that species' traits can be a compromise among many functions [74]. The beak's origin as a modified jaw [75] strongly suggests that the foraging function of the beak predates thermoregulation, nest building, preening, etc. However, once these other exaptive functions evolved [76], changes in beak shape and size would have pleiotropic consequences beyond foraging efficiency (as acknowledged in [77]). This implies a one-to-many mapping of trait form to function that should engender numerous trade-offs, unless these are reduced by the mapping of many potential trait optima to each function as in [78].

The size of honeyeater species' beaks not only increased in warmer climates but also increased with their reliance on nectar foraging behaviour; beaks also become more elongate under these conditions. This is likely to reflect a set of conflicting selection pressures similar to that observed in studies of granivorous song sparrows [19,79], wherein individuals with short beaks exhibited lower over-winter survival as juveniles, but greater reproductive success as adults. This should produce evolutionary trade-offs in some cases, especially for honeyeaters that forage on nectar but must endure relatively

cold winters. Other behaviours may also mitigate the costs of a large beak in cold weather, such as the tendency of many bird species to tuck their beak into their feathers [80,81]. Likewise, honeyeaters are observed to perform a diverse range of foraging behaviours that could mitigate the costs of adaptations to extreme climates [41]. We propose that the inability to simultaneously maximize multiple functions of a single trait [79] may lead to the evolution of novel behaviours either solving or ameliorating the dilemma.

## (c) Consequences for song evolution

Variation in both the beak size and beak shape had a significant effect on song characteristics in honeyeaters (figure 4). Larger honeyeater species sang at lower frequencies, reiterating a relationship that is well established both within and among species [49,82]. Species with longer and narrower beaks also sang at lower maximum frequencies, and species with larger beaks sang at a slower pace. Longer beaks have been observed to attenuate high-frequency sounds produced by the syrinx in laboratory experiments [83,84]. Likewise, a larger beak should create performance constraints on the production of rapid vocal tract movements [10,14]. This effect may be especially pronounced in honeyeaters, where we observe that species with large beaks tend to use them for nectarivory and hawking, rather than for more mechanically demanding behaviours like seed-crushing [13]. Assuming the beak to be a simple lever, this motor performance relationship between the beak size and the song characteristics should be mediated by limitations not only in the beak's mass and length but also in the muscles that move it—their mass, length and insertion points [85]. Future studies are needed that can compare variation in musculature and beak morphology among species [68,86].

Habitat structure is often implicated as an influence on vocal evolution in birds [87,88]. The absence of this variable in our analysis is a limitation of this study, as it could potentially have complex and confounding relationships with foraging ecology and the Australian climate. However, recent studies suggest that while direct selection on song phenotypes based on habitat is likely to be important among speciating populations [89,90], such effects can be weak or outweighed by indirect trait relationships during diversification [30,82].

Previous studies uncovered the indirect link between foraging ecology and song performance by examining species known to or expected to vary in the beak size based on their foraging ecology [10,14,16,91]. Our study adds to this background by connecting beak function to beak morphology to vocal evolution in a single phylogenetic comparative framework. In doing so, we show how changes in beak morphology associated with both foraging and thermoregulation can influence vocal evolution in songbirds, although we cannot rule out a reverse effect wherein changes in song drive changes in beak morphology. Divergence in trait function can be expected to cause signal divergence during speciation [31]. Also, environment–trait–signal relationships like the one we report could potentially encode information on functional trait genotype into signals, which could form the basis of assortative mating [92].

## 5. Conclusion

Here, we assess the influence of foraging behaviour and climate on the evolution of beak morphology and find that both are correlated with changes in beak size and shape. Indeed, no axis of morphological variation that we examined was associated with an isolated function. The at times conflicting effects of variation in this trait have been observed in wild populations [19]; here, we show that they can also be observed when comparing lineages in a broadly diversifying continental radiation. Our findings add to those of previous comparative studies [18,64] by showing that the thermoregulatory and foraging functions they identified are correlated with changes in beak morphology. Many traits typically associated with foraging exhibit a degree of multifunctionality similar to the avian beak, which produces a one-to-many mapping of form to function [78]. For example, the mandibles of worker ants are associated with a broad behavioural repertoire [93,94], as are the pedipalps of many arachnids [95]. Such multifunctional traits may occasionally appear highly specialized for a single task, but it seems likely that this specialization comes at a cost for performance of other tasks [5,27].

Our study confirms others that find relationships between beak size and song evolution [10,14,16]. Our results show that the elongation of the beak may also have implications for vocal performance and further add to these previous works by connecting foraging and thermoregulatory functions to their signalling consequences. In particular, honeyeaters evolving beak shapes that are associated with nectarivory should be limited in the maximum frequency that their songs can achieve. Similarly, species with small beaks associated with winter cold tolerance should be capable of producing faster songs. Divergent song characteristics may contribute to the evolution of reproductive isolating barriers [31] or potentially provide information to receivers about the signaller's functional phenotype [92]. We predict that signal characteristics associated with functional phenotypes that convey a selective advantage should thus be ideal targets for the evolution of female preferences and assortative mating, as they may convey information regarding a suitor's fitness without respect to the condition. Taken together, the evolution of the avian beak exemplifies how morphological traits can represent an evolutionary compromise among functions, with downstream consequences for behaviour and communication.

Data accessibility. Trait data have been uploaded to the Dryad Digital Repository: https://doi.org/10.5061/dryad.crjdfn312 [96]. All measurements were taken from the vouchered specimens and recordings indicated in the electronic supplementary material, appendices S1 and S2.

Authors' contributions. N.R.F. designed the study, collected morphological and song measurements, performed the analyses and drafted the manuscript; E.T.M. contributed behavioural data and field observations; J.R.B. and H.K. collected and processed song measurements; this research was co-supervised by V.R. and E.P.E., who contributed equally to study design and editing. All authors gave final approval for publication and agree to be held accountable for work therein.

Competing interests. We declare we have no competing interests

Funding. This study was supported by the Grant Agency of the Czech Republic (Project no. 16-22379S). J.R.B., E.P.E., N.R.F. and H.K. were supported by subsidy funding to OIST, and by Japan Society for the Promotion of Science KAKENHI grants (grant nos. 17K15180 to E.P.E. and 17K15178 to N.R.F., respectively).

Acknowledgements. We thank Mark Adams, the curators and staff of the Natural History Museum at Tring, and especially the Scharsachs for their hospitality during N.R.F.'s extended visit to the collections. We also thank Alex Drew and Leo Joseph at ANWC, and Matthew Young at the Macaulay Library for their assistance. This work

would not be possible without the dedicated efforts of many recordists and natural historians (see the electronic supplementary material, appendix S2). We appreciate thoughtful suggestions from Dan Bolnick, Jen Bright, Chris Cooney, Emma Greig, Jeff Podos, Peter Wainwright and Dan Warren.

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
