## [Reviewer comments · Proceedings of the Royal Society B: Biological Sciences]

Review History

RSPB-2019-0498.R0 (Original submission)

Review form: Reviewer 1

Recommendation

Major revision is needed (please make suggestions in comments)

Scientific importance: Is the manuscript an original and important contribution to its field?
Good

General interest: Is the paper of sufficient general interest?
Excellent

Quality of the paper: Is the overall quality of the paper suitable?
Good

Is the length of the paper justified?
Yes

Should the paper be seen by a specialist statistical reviewer?

No

Do you have any concerns about statistical analyses in this paper? If so, please specify them explicitly in your report.

No

It is a condition of publication that authors make their supporting data, code and materials available - either as supplementary material or hosted in an external repository. Please rate, if applicable, the supporting data on the following criteria.

Is it accessible?

N/A

Is it clear?

N/A

Is it adequate?

N/A

Do you have any ethical concerns with this paper?

No

Comments to the Author

Overall I think this is a great study that combined extensive datasets from different origins to explore very interesting questions regarding the evolution of beaks in birds. The authors compare the role of different selective pressures in the evolution of beak shape in honeyeaters (Meliphagidae) and find that foraging ecology and climate affect differently two aspects of beak morphology.

I think the paper is written very clearly, and the authors have done a great job in condensing and summarising a great amount of information to make it accessible to the reader. I only have one issue with that, and is that the statistical analyses seem to be too simplified and there could be details missing that some readers would like to have, in case someone wants to do similar analyses. I hope the authors find the following comments useful:

1. It wasn't clear to me what type of model they used for the multivariate analyses, they cite Adams (2014), was geomorph used? If so, would be good to specify. Also good to mention explicitly which were the predictor variables used, because there are different sets of predictors in the path analysis described previously.
2. It wasn't clear to me either how body size was controlled for in the path analyses. The authors mention that in PGLS analyses body and beak size were not included in the same model because of collinearity, I imagine the same logic was applied to the path analyses. However, would this mean that the effects of temperature on beak size reported in the path analyses are actually effects on body size? Would it be possible to use the residuals of beak size (controlled by body size) as variable in this analysis? I guess it becomes a question of whether climate/nectarivory affect relative or absolute beak size.
3. I understand the logic in arguing that multifunctional trait model has a higher fit than single function models (according to the path analysis). Maybe it would be good to clarify that the statistic used for comparison (CICc) takes into account the number of paths, and models with

more paths are not always favoured (e.g. in general we would expect a model with more parameters to explain more variation than a model with less parameters (e.g. paths)).

4. The authors suggest that nectarivory contributes more than climate in driving the evolution of beak shape (curvature and depth). However, the path analysis shows that climate can significantly affect nectarivory, and so climate could have an indirect effect on beak shape through nectarivory. It would be interesting to calculate, compare and discuss these indirect routes, to me this is one of the interesting features of path analyses. Also, in figure 2, are nectar, curvature, depth different PCs (those in Fig 2C)? If so, it would be good to clarify that in text and in the fig (e.g. Nectar PC1).

5. It would be nice to see a stronger connection between the analyses on song and the other analyses. This is mentioned briefly on lines 330-335 but I think the point could be stronger. Maybe it's too much, but as a suggestion, the authors could make a figure combining the findings from all the analyses. Like a simplified path graph showing how climate/foraging variables affect which beak traits and which beak traits affect which song traits. I think this is one of the most interesting points of this study and could be made stronger. I also think that figures 1a and 4a are not totally necessary and could be moved to supplementary material in case the authors want more space.

Minor comments:

Line 26: I think the 'while' could be replaced with a 'however' after the comma.

Line 118: How many species?

Figure 1: The picture looks like it is on an angle and it is not a standard one (e.g. 90), which made me wonder how the authors could consistently take pictures from that same angle (e.g. slightly from the side). Is this an actual picture used? If so, maybe clarify where pictures were taken from and whether the angle was consistent.

Line 217: It would be good to present a table in the supplementary material with the specific results of the PGLS models (e.g. Beta, T-value, P value).

Line 255: I don't understand how it suggests that selection on these functions is evident during diversification, please clarify.

Overall it was a very nicely written and interesting paper, congratulations.

Review form: Reviewer 2

Recommendation

Accept with minor revision (please list in comments)

Scientific importance: Is the manuscript an original and important contribution to its field?

Excellent

General interest: Is the paper of sufficient general interest?

Excellent

Quality of the paper: Is the overall quality of the paper suitable?

Excellent

Is the length of the paper justified?

Yes

Should the paper be seen by a specialist statistical reviewer?

No

Do you have any concerns about statistical analyses in this paper? If so, please specify them explicitly in your report.

Yes

It is a condition of publication that authors make their supporting data, code and materials available - either as supplementary material or hosted in an external repository. Please rate, if applicable, the supporting data on the following criteria.

Is it accessible?

No

Is it clear?

No

Is it adequate?

No

Do you have any ethical concerns with this paper?

No

Comments to the Author

I have reviewed the manuscript entitled "Evolution of a multifunctional trait: shared effects of foraging ecology and thermoregulation on beak morphology, with consequences for song evolution" and I find it to be a sound and interesting paper. The premise of examining many predictors of beak shape is rare, as the authors present in the thorough literature appraisal of the introduction. The paper is well written and enjoyable to read. The data are impressive and sufficient to answer the question, and the analytics are mostly appropriate - with one issue detailed below that will possibly change the outcomes.

My main concern regards the treatment of the foraging behaviour data. A 'phylogenetic' PCA is not appropriate here because this serves to "double-correct" the data when coupled with a PGLS. You have not done a phylogenetic-PCA for the shape data, so why do so for the diet data? Phylogenetic PCAs are often mistreated in analyses; since PCA is not a statistical approach that assumes independence of the input variables, it does not need to be explicitly altered to account for the statistical non-independence of species data. A standard PCA would suffice for reducing dimensionality of the diet data if it was strictly necessary to do so.

Since foraging behaviour is a matrix of variables considered together, as is the Procrustes residuals of shape data, they are suitable for multivariate methods. As a complement to the path analysis, I suggest the authors consider the two-block partial least squares analysis (2B-PLS) to consider the patterns of covariation between the different factors and beak shape. For example, one can examine beak shape (one 'block', aka matrix, of the Procrustes residuals) and foraging behaviour data (second block, the diet matrix), and show which behavioural traits influence specific shape changes. This is a standard technique for looking for axes of covariation between two multivariate datasets (Rohlf FJ & Corti M, 2000. The use of partial least-squares to study covariation in shape. *Systematic Biology*), and this can be done in a phylogenetic context (Adams, D.C. & Felice, R.N., 2014. Assessing Trait Covariation and Morphological Integration on Phylogenies Using Evolutionary Covariance Matrices. *PLoS ONE*) and implementable in the same R package used, geomorph. The strength of covariation can then be compared to

covariation of shape with other multidimensional blocks (song characteristics, climate) through multiple 2B-PLS analyses, and using an approach that is presented as a way to compare patterns of morphological integration (Adams, D.C. & Collyer, M.L., 2016. On the comparison of the strength of morphological integration across morphometric datasets. Evolution), but that simply serves as a method to compare covariation coefficients across different comparisons in any type of data.

I see that the authors mention “multidimensional partial least squares analysis” in the caption of Figure 3 (line 657), but with the wrong citation (55, that for multidimensional PGLS). But with no mention of the PLS in the methods, I am not at sure what they have done. This needs to be clarified also.

The authors seem to swap between using the whole shape data matrix, and the PC axes on their own in statistical analysis, which is confusing to follow. Please change the sentence Line 127-128, because they are not always being used in subsequent analyses, and be explicit where these PC axes are being used as separate variables, and where the Procrustes residuals shape matrix is being used.

Only the upper part of the beak was measured in this study. I assume the lower part is not standard in shape or size, and therefore the study is missing an important aspect of the overall beak shape. A comment on how this may affect the results and interpretations would be good to see.

Line 127 – Principal not principle

Figure 1 – PC axes must have values on the tick marks. Would also be good to have % of variance on each PC axis label.

Review form: Reviewer 3 (Matthew R.E. Symonds)

Recommendation

Major revision is needed (please make suggestions in comments)

Scientific importance: Is the manuscript an original and important contribution to its field?

Good

General interest: Is the paper of sufficient general interest?

Good

Quality of the paper: Is the overall quality of the paper suitable?

Marginal

Is the length of the paper justified?

Yes

Should the paper be seen by a specialist statistical reviewer?

No

Do you have any concerns about statistical analyses in this paper? If so, please specify them explicitly in your report.

Yes

It is a condition of publication that authors make their supporting data, code and materials available - either as supplementary material or hosted in an external repository. Please rate, if applicable, the supporting data on the following criteria.

Is it accessible?

No

Is it clear?

N/A

Is it adequate?

N/A

Do you have any ethical concerns with this paper?

No

Comments to the Author

In this paper, as the title indicates, the authors demonstrate how foraging ecology and temperature influence elements of beak morphology using data from Australasian honeyeaters. On the plus side, it's really nice to see a paper consider BOTH diet/foraging strategy and thermoregulation and their relative influence on beak morphology. This is good because, in my experience, many ornithologists consider the role of temperature in driving changes in beak size to be miniscule in comparison with foraging ecology. Therefore, it's important to see a paper demonstrate this not to be the case. I find the authors main conclusions basically convincing. The addition of the analysis investigating the effects of beak morphology on song is interesting.

However, the paper at present does have quite a number of issues in regard to the methods and results/interpretation which need to be addressed, which I found dampened my enthusiasm for the study. I think (or hope) that these concerns can be addressed through some more careful explanation/wording, but because of the lack of clarity I can't entirely rule out that the analyses and interpretation have some more fundamental flaws.

Major concerns:

The paper makes clear how elements of beak shape are estimated (through PCs), but is very unclear as to how beak size is actually calculated, despite this being a major part of the paper. The only mention of beak size in the methods is in the final few lines, in relation to analysing its effect on song structure, but with no detail of how it is measured. Also since the beak measurements appear to only consider a proportion of the upper beak, I wonder to what extent beak size has been accurately estimated (although without further detail it's impossible to know if this is a fair criticism). On a more minor level, this issue of how beak size is actually defined also affects the introduction in lines 71-73, where individual elements beak depth and beak length, are described as if they are independent of beak size (I'd suggest simply describing these measures as 'elements of beak shape', which they also are).

The results present some difficulties in interpretation, in part because two different approaches are used - phylogenetic path analysis, and then, in order to generate pseudo-R2 values, a multivariate phylogenetic regression (by which I assume it to mean that 'shape' generally is the response variable, rather than the individual components of shape (elongation and curvature - though the former is confusingly called 'Depth' in Figure 2e). One problem here is that the effects on beak size are not clearly spelt out - merely that climate is a stronger predictor (as judged from Figure 3a) than foraging ecology. There is one mention (line 308) that the size of beaks increases

with warmer temperatures. However, Figure 2e presents a confusing picture in that winter temperature is positively associated with beak size (i.e. warmer climates = bigger beaks), but summer temperatures are negatively associated (i.e. warmer climates = smaller beaks). I appreciate that in part this has been more explicitly investigated in the authors' previous paper in *Evolution* – where winter temperature is a better predictor than summer temperature. However, the confusing thing here is that Figure 2e indicates that the effects of winter and summer temperatures are both significant, but Fig 3a indicates that only winter temperature has a significant effect. However, in this case why is there a discrepancy between the two analyses. I think the problem here may lie in not having explained how the measure of beak size is generated – but in any event more discussion is needed in the text of the way in which climate actually influences beak size (see a bit more discussion below). There is also some confusion when comparing the results of Figure 2e and Figure 3a, in linking the foraging ecology PCs with 'Nectar'. To me it seems like the model on which the analysis in Figure 3a is based is different from the model in figure 2e – in which case shouldn't more details of the model in 3a be given, rather than just R2 values?

On a related note at lines 234-235 it is stated that the effect of summer temperature on shape is greater than that of winter temperature – presumably based on the results in Fig 3b, however, this does not appear to be supported by Fig 2e, where the correlation coefficients for summer temperature with shape elements is smaller in magnitude than that link winter temperature to depth. I think I know what the answer to this is (you are referring in 3b to shape overall, rather than elements of shape) but I still think this should be explicitly discussed.

Regarding the contradiction between the effects of summer temperature and winter temperature, might the negative (or lack of significant relationship) with the former be explained by very hot environments causing selection for SMALLER beaks due to them becoming a liability in taking on heat when ambient temperatures elevate beyond the bird body temperature. Greenberg & Danner (2012) *Evolution* demonstrated how curvilinear relationships of beak size with summer temperature can result from this. It would be interesting to either examine whether there is evidence for any curvilinear relationship, or at least get some idea of the range of maximum summer temperatures experienced by the species in the analysis.

Line 122 – two questions raised here: the measurement area of the beak is not entirely clear – did you just examine the area of the upper beak anterior to the nares? And also how did you align the angle of the beak towards the camera so that this was consistent - this would surely affect the judgement on the landmarks?

Line 180 – how did you deal with intraspecific variation (or at least species sample size) in the analysis? This is something that should be taken into consideration (see chapter by Garamszegi in the book on *Modern Phylogenetic Comparative Methods*).

I was a bit concerned about how the authors dealt with body size as a controlling factor in the analysis of song evolution. Because of the high correlation between beak size and body size they did not include both simultaneously in any of the PGLS analyses. For one thing, it's not clear how it was decided 'a priori' (line 200) which was the more relevant.

Additionally, the issue here is that it's not therefore possible to evaluate to what extent it is body size or beak size that is driving song evolution. Because using residuals values is not appropriate (see Freckleton 2002 *J Anim Ecol*), it might be suggested to still include both variables in the model as suggested by Freckleton. If the authors don't feel this is valid (I must admit I've not seen this issue addressed specifically in response to Freckleton's paper), then at least I'd suggest comparing models including either body size or beak size and see which provides a better fit for the data.

Line 154-156 – this last sentence is not clear (the e.g. in brackets does not explain what you are getting at here).

Can you be certain that beak shape/size drives song evolution and it's not the other way round (i.e. selection on song is driving changes in beak shape/size) – some reference here as to why it's definitely the other way round would be good.

Lines 211-213 – this analysis of phylogenetic signal is not covered in the methods, nor is the reason for it justified/explained.

Minor things:

Line 27: a very minor thing – but I would write this as more simply as Bird beaks, rather than Birds' beaks

Line 59: when referring to preening you could also mention that beak shape is related to efficient parasite removal – see Villa et al. (2018) *Evolutionary Ecology* 32: 443-452

Line 60: there is a more recent extensive review of the role of the avian beak in thermoregulation which would be appropriate to cite here: Tattersall et al. (2017) *Biological Reviews* 92: 1630-1656

Lines 76-78: You should also probably discuss somewhere the new paper from the same group, which finds complex relationships between beak shape and feeding ecology - Navalon et al. (2019) *Evolution* 73: 422-435

Line 98: I don't know what you mean by "a conveniently controlled replicate of the evolutionary experiment". I'd drop this expression.

Line 101: It may be worth pointing out somewhere that Gardner et al (2016) *Climate Change Responses* 3: 11 found no direct link between temperature and bill size in Meliphagoidea, but did find that temperature mediates the effect of humidity on bill size - in general it would be worth including some (brief) discussion later of the effect of humidity on bill size.

Methods section a) this seems curiously placed here right at the start of the methods. I would include the information about phylogeny in the later section on comparative methods where it is more directly relevant.

Line 181 – it would be to explain what lambda is for those readers who do not know.

Lines 257-260 - this seems an odd comparison/statement. implies that comparing to the Galapagos finches was intended as a key aim of the study in the first place. The statement seems a bit redundant (even if is correct).

Line 438 – reference 17 doesn't seem like it is correctly given (missing volume number and real page numbers?)

Line 518 – there is a little 'a' after Tobias J – presumably this should be Tobias JA.

Line 664 – Standardised beta values - Does this mean the variables were standardized prior to analysis, or that you standardised after calculation of the beta values (if so, how? This information should be given in the methods).

Decision letter (RSPB-2019-0498.R0)

22-Mar-2019

Dear Dr Friedman:

I am writing to inform you that your manuscript RSPB-2019-0498 entitled "Evolution of a multifunctional trait: shared effects of foraging ecology and thermoregulation on beak morphology, with consequences for song evolution" has, in its current form, been rejected for publication in Proceedings B.

This action has been taken on the advice of referees, who have recommended that substantial revisions are necessary. With this in mind we would be happy to consider a resubmission, provided the comments of the referees are fully addressed. However please note that this is not a provisional acceptance.

Sincerely,
Proceedings B
<mailto:proceedingsb@royalsociety.org>

Associate Editor
Board Member: 1
Comments to Author:

This study shows that different selection pressures drive different aspects of beak morphology (specialisation along different functional axes): climate has strongest influence on beak size, foraging ecology has strongest influence on beak shape and beak shape is also correlated with vocal performance (song characteristics). A strength of the study is that it assesses multiple selection pressures/potential functions whereas most studies test a single function for a given morphological feature. All three reviewers agree that the study would be of broad interest to readers of PRSLB. However, all three reviewers raised some serious methodological concerns that would need to be addressed before the paper could be published. In particular, clarification is

needed on how beak size was measured, and how body size was controlled for in the path analysis. Reviewer 1 also has some good suggestions for aspects that could be developed further, particularly in relation to beak shape and song. If these methodological issues are addressed the study will make a strong contribution to our understanding of morphological evolution.

Reviewer(s)' Comments to Author:

Referee: 1

Comments to the Author(s)

Overall I think this is a great study that combined extensive datasets from different origins to explore very interesting questions regarding the evolution of beaks in birds. The authors compare the role of different selective pressures in the evolution of beak shape in honeyeaters (Meliphagidae) and find that foraging ecology and climate affect differently two aspects of beak morphology.

I think the paper is written very clearly, and the authors have done a great job in condensing and summarising a great amount of information to make it accessible to the reader. I only have one issue with that, and is that the statistical analyses seem to be too simplified and there could be details missing that some readers would like to have, in case someone wants to do similar analyses. I hope the authors find the following comments useful:

1. It wasn't clear to me what type of model they used for the multivariate analyses, they cite Adams (2014), was geomorph used? If so, would be good to specify. Also good to mention explicitly which were the predictor variables used, because there are different sets of predictors in the path analysis described previously.
2. It wasn't clear to me either how body size was controlled for in the path analyses. The authors mention that in PGLS analyses body and beak size were not included in the same model because of collinearity, I imagine the same logic was applied to the path analyses. However, would this mean that the effects of temperature on beak size reported in the path analyses are actually effects on body size? Would it be possible to use the residuals of beak size (controlled by body size) as variable in this analysis? I guess it becomes a question of whether climate/nectarivory affect relative or absolute beak size.
3. I understand the logic in arguing that multifunctional trait model has a higher fit than single function models (according to the path analysis). Maybe it would be good to clarify that the statistic used for comparison (CICc) takes into account the number of paths, and models with more paths are not always favoured (e.g. in general we would expect a model with more parameters to explain more variation than a model with less parameters (e.g. paths)).
4. The authors suggest that nectarivory contributes more than climate in driving the evolution of beak shape (curvature and depth). However, the path analysis shows that climate can significantly affect nectarivory, and so climate could have an indirect effect on beak shape through nectarivory. It would be interesting to calculate, compare and discuss these indirect routes, to me this is one of the interesting features of path analyses. Also, in figure 2, are nectar, curvature, depth different PCs (those in Fig 2C)? If so, it would be good to clarify that in text and in the fig (e.g. Nectar PC1).
5. It would be nice to see a stronger connection between the analyses on song and the other analyses. This is mentioned briefly on lines 330-335 but I think the point could be stronger. Maybe it's too much, but as a suggestion, the authors could make a figure combining the findings from

all the analyses. Like a simplified path graph showing how climate/foraging variables affect which beak traits and which beak traits affect which song traits. I think this is one of the most interesting points of this study and could be made stronger. I also think that figures 1a and 4a are not totally necessary and could be moved to supplementary material in case the authors want more space.

Minor comments:

Line 26: I think the 'while' could be replaced with a 'however' after the comma.

Line 118: How many species?

Figure 1: The picture looks like it is on an angle and it is not a standard one (e.g. 90), which made me wonder how the authors could consistently take pictures from that same angle (e.g. slightly from the side). Is this an actual picture used? If so, maybe clarify where pictures were taken from and whether the angle was consistent.

Line 217: It would be good to present a table in the supplementary material with the specific results of the PGLS models (e.g. Beta, T-value, P value).

Line 255: I don't understand how it suggests that selection on these functions is evident during diversification, please clarify.

Overall it was a very nicely written and interesting paper, congratulations.

Referee: 2

Comments to the Author(s)

I have reviewed the manuscript entitled "Evolution of a multifunctional trait: shared effects of foraging ecology and thermoregulation on beak morphology, with consequences for song evolution" and I find it to be a sound and interesting paper. The premise of examining many predictors of beak shape is rare, as the authors present in the thorough literature appraisal of the introduction. The paper is well written and enjoyable to read. The data are impressive and sufficient to answer the question, and the analytics are mostly appropriate – with one issue detailed below that will possibly change the outcomes.

My main concern regards the treatment of the foraging behaviour data. A 'phylogenetic' PCA is not appropriate here because this serves to "double-correct" the data when coupled with a PGLS. You have not done a phylogenetic-PCA for the shape data, so why do so for the diet data? Phylogenetic PCAs are often mistreated in analyses; since PCA is not a statistical approach that assumes independence of the input variables, it does not need to be explicitly altered to account for the statistical non-independence of species data. A standard PCA would suffice for reducing dimensionality of the diet data if it was strictly necessary to do so.

Since foraging behaviour is a matrix of variables considered together, as is the Procrustes residuals of shape data, they are suitable for multivariate methods. As a complement to the path analysis, I suggest the authors consider the two-block partial least squares analysis (2B-PLS) to consider the patterns of covariation between the different factors and beak shape. For example, one can examine beak shape (one 'block', aka matrix, of the Procrustes residuals) and foraging behaviour data (second block, the diet matrix), and show which behavioural traits influence specific shape changes. This is a standard technique for looking for axes of covariation between two multivariate datasets (Rohlf FJ & Corti M, 2000. The use of partial least-squares to study covariation in shape. *Systematic Biology*), and this can be done in a phylogenetic context (Adams, D.C. & Felice, R.N., 2014. Assessing Trait Covariation and Morphological Integration on Phylogenies Using Evolutionary Covariance Matrices. *PLoS ONE*) and implementable in the same R package used, geomorph. The strength of covariation can then be compared to

covariation of shape with other multidimensional blocks (song characteristics, climate) through multiple 2B-PLS analyses, and using an approach that is presented as a way to compare patterns of morphological integration (Adams, D.C. & Collyer, M.L., 2016. On the comparison of the strength of morphological integration across morphometric datasets. *Evolution*), but that simply serves as a method to compare covariation coefficients across different comparisons in any type of data.

I see that the authors mention “multidimensional partial least squares analysis” in the caption of Figure 3 (line 657), but with the wrong citation (55, that for multidimensional PGLS). But with no mention of the PLS in the methods, I am not at sure what they have done. This needs to be clarified also.

The authors seem to swap between using the whole shape data matrix, and the PC axes on their own in statistical analysis, which is confusing to follow. Please change the sentence Line 127-128, because they are not always being used in subsequent analyses, and be explicit where these PC axes are being used as separate variables, and where the Procrustes residuals shape matrix is being used.

Only the upper part of the beak was measured in this study. I assume the lower part is not standard in shape or size, and therefore the study is missing an important aspect of the overall beak shape. A comment on how this may affect the results and interpretations would be good to see.

Line 127 – Principal not principle

Figure 1 – PC axes must have values on the tick marks. Would also be good to have % of variance on each PC axis label.

Referee: 3

Comments to the Author(s)

In this paper, as the title indicates, the authors demonstrate how foraging ecology and temperature influence elements of beak morphology using data from Australasian honeyeaters. On the plus side, it’s really nice to see a paper consider BOTH diet/foraging strategy and thermoregulation and their relative influence on beak morphology. This is good because, in my experience, many ornithologists consider the role of temperature in driving changes in beak size to be miniscule in comparison with foraging ecology. Therefore, it’s important to see a paper demonstrate this not to be the case. I find the authors main conclusions basically convincing. The addition of the analysis investigating the effects of beak morphology on song is interesting.

However, the paper at present does have quite a number of issues in regard to the methods and results/interpretation which need to be addressed, which I found dampened my enthusiasm for the study. I think (or hope) that these concerns can be addressed through some more careful explanation/wording, but because of the lack of clarity I can’t entirely rule out that the analyses and interpretation have some more fundamental flaws.

Major concerns:

The paper makes clear how elements of beak shape are estimated (through PCs), but is very unclear as to how beak size is actually calculated, despite this being a major part of the paper. The only mention of beak size in the methods is in the final few lines, in relation to analysing its effect on song structure, but with no detail of how it is measured. Also since the beak measurements

appear to only consider a proportion of the upper beak, I wonder to what extent beak size has been accurately estimated (although without further detail it's impossible to know if this is a fair criticism). On a more minor level, this issue of how beak size is actually defined also affects the introduction in lines 71-73, where individual elements beak depth and beak length, are described as if they are independent of beak size (I'd suggest simply describing these measures as 'elements of beak shape', which they also are).

The results present some difficulties in interpretation, in part because two different approaches are used – phylogenetic path analysis, and then, in order to generate pseudo-R² values, a multivariate phylogenetic regression (by which I assume it to mean that 'shape' generally is the response variable, rather than the individual components of shape (elongation and curvature – though the former is confusingly called 'Depth' in Figure 2e). One problem here is that the effects on beak size are not clearly spelt out – merely that climate is a stronger predictor (as judged from Figure 3a) than foraging ecology. There is one mention (line 308) that the size of beaks increases with warmer temperatures. However, Figure 2e presents a confusing picture in that winter temperature is positively associated with beak size (i.e. warmer climates = bigger beaks), but summer temperatures are negatively associated (i.e. warmer climates = smaller beaks). I appreciate that in part this has been more explicitly investigated in the authors' previous paper in *Evolution* – where winter temperature is a better predictor than summer temperature. However, the confusing thing here is that Figure 2e indicates that the effects of winter and summer temperatures are both significant, but Fig 3a indicates that only winter temperature has a significant effect. However, in this case why is there a discrepancy between the two analyses. I think the problem here may lie in not having explained how the measure of beak size is generated – but in any event more discussion is needed in the text of the way in which climate actually influences beak size (see a bit more discussion below). There is also some confusion when comparing the results of Figure 2e and Figure 3a, in linking the foraging ecology PCs with 'Nectar'. To me it seems like the model on which the analysis of Figure 3a is based is different from the model in figure 2e – in which case shouldn't more details of the model in 3a be given, rather than just R² values?

On a related note at lines 234-235 it is stated that the effect of summer temperature on shape is greater than that of winter temperature – presumably based on the results in Fig 3b, however, this does not appear to be supported by Fig 2e, where the correlation coefficients for summer temperature with shape elements is smaller in magnitude than that link winter temperature to depth. I think I know what the answer to this is (you are referring in 3b to shape overall, rather than elements of shape) but I still think this should be explicitly discussed.

Regarding the contradiction between the effects of summer temperature and winter temperature, might the negative (or lack of significant relationship) with the former be explained by very hot environments causing selection for SMALLER beaks due to them becoming a liability in taking on heat when ambient temperatures elevate beyond the bird body temperature. Greenberg & Danner (2012) *Evolution* demonstrated how curvilinear relationships of beak size with summer temperature can result from this. It would be interesting to either examine whether there is evidence for any curvilinear relationship, or at least get some idea of the range of maximum summer temperatures experienced by the species in the analysis.

Line 122 – two questions raised here: the measurement area of the beak is not entirely clear – did you just examine the area of the upper beak anterior to the nares? And also how did you align the angle of the beak towards the camera so that this was consistent - this would surely affect the judgement on the landmarks?

Line 180 – how did you deal with intraspecific variation (or at least species sample size) in the

analysis? This is something that should be taken into consideration (see chapter by Garamszegi in the book on Modern Phylogenetic Comparative Methods).

I was a bit concerned about how the authors dealt with body size as a controlling factor in the analysis of song evolution. Because of the high correlation between beak size and body size they did not include both simultaneously in any of the PGLS analyses. For one thing, it's not clear how it was decided 'a priori' (line 200) which was the more relevant.

Additionally, the issue here is that it's not therefore possible to evaluate to what extent it is body size or beak size that is driving song evolution. Because using residuals values is not appropriate (see Freckleton 2002 *J Anim Ecol*), it might be suggested to still include both variable in the model as suggested by Freckleton. If the authors don't feel this is valid (I must admit I've not seen this issue addressed specifically in response to Freckleton's paper), then at least I'd suggest comparing models including either body size or beak size and see which provides a better fit for the data.

Line 154-156 – this last sentence is not clear (the e.g. in brackets does not explain what you are getting at here).

Can you be certain that beak shape/size drives song evolution and it's not the other way round (i.e. selection on song is driving changes in beak shape/size) – some reference here as to why it's definitely the other way round would be good.

Lines 211-213 – this analysis of phylogenetic signal is not covered in the methods, nor is the reason for it justified/explained.

Minor things:

Line 27: a very minor thing – but I would write this as more simply as Bird beaks, rather than Birds' beaks

Line 59: when referring to preening you could also mention that beak shape is related to efficient parasite removal – see Villa et al. (2018) *Evolutionary Ecology* 32: 443-452

Line 60: there is a more recent extensive review of the role of the avian beak in thermoregulation which would be appropriate to cite here: Tattersall et al. (2017) *Biological Reviews* 92: 1630-1656

Lines 76-78: You should also probably discuss somewhere the new paper from the same group, which finds complex relationships between beak shape and feeding ecology - Navalon et al. (2019) *Evolution* 73: 422-435

Line 98: I don't know what you mean by "a conveniently controlled replicate of the evolutionary experiment". I'd drop this expression.

Line 101: It may be worth pointing out somewhere that Gardner et al (2016) *Climate Change Responses* 3: 11 found no direct link between temperature and bill size in Meliphagoidea, but did find that temperature mediates the effect of humidity on bill size - in general it would be worth including some (brief) discussion later of the effect of humidity on bill size.

Methods section a) this seems curiously placed here right at the start of the methods. I would include the information about phylogeny in the later section on comparative methods where it is more directly relevant.

Line 181 – it would be to explain what lambda is for those readers who do not know.

Lines 257-260 - this seems an odd comparison/statement. implies that comparing to the Galapagos finches was intended as a key aim of the study in the first place. The statement seems a bit redundant (even if is correct).

Line 438 - reference 17 doesn't seem like it is correctly given (missing volume number and real page numbers?)

Line 518 - there is a little 'a' after Tobias J - presumably this should be Tobias JA.

Line 664 - Standardised beta values - Does this mean the variables were standardized prior to analysis, or that you standardised after calculation of the beta values (if so, how? This information should be given in the methods).

Author's Response to Decision Letter for (RSPB-2019-0498.R0)

See Appendix A.

RSPB-2019-2474.R0

Review form: Reviewer 1

Recommendation

Accept with minor revision (please list in comments)

Scientific importance: Is the manuscript an original and important contribution to its field?

Good

General interest: Is the paper of sufficient general interest?

Good

Quality of the paper: Is the overall quality of the paper suitable?

Good

Is the length of the paper justified?

Yes

Should the paper be seen by a specialist statistical reviewer?

No

Do you have any concerns about statistical analyses in this paper? If so, please specify them explicitly in your report.

No

It is a condition of publication that authors make their supporting data, code and materials available - either as supplementary material or hosted in an external repository. Please rate, if applicable, the supporting data on the following criteria.

Is it accessible?

Yes

Is it clear?

Yes

Is it adequate?

Yes

Do you have any ethical concerns with this paper?

No

Comments to the Author

I appreciate the effort that the authors have made to follow our suggestions in the previous round of reviews. The analyses are explained much more clearly and I am satisfied with most of the responses to my comments and suggestions.

After reading again the manuscript I have a few more comments to add:

Line 87: I am not sure if this whole 'speciation' framework is needed, or adds much... the current study is far from informing on the divergence aspect of things. I think is already interesting enough with just saying that it can affect species interactions by affecting song traits. Just a thought...

Line 90: But they are oscines as well, the sentence sounds confusing, maybe reword?

Line 101: 'Song behaviour' feels strange, maybe song traits or characteristics?

Line 150: 'NaturalisT'?

Line 204: I was confused by the number and thought there was a mistake. Maybe better to not to begin the paragraph with the number.

Line 212: In the text and revisions the authors mention they ended up using residuals too but I couldn't find the results from that analysis. Given that reviewer 3 had concerns regarding the use of residuals maybe best to leave out? Please update properly.

Line 227: Please explain what the PLS analysis will be used for, what will it test?

Line 243: Is this PC1 of elongation the same PC of 'Depth' in the path analysis figure? If so please name the same way (and add in the figure that it is a PC). I still find confusing why they are called foraging PCs or shape PCs in some tables but just 'Nectar', 'Depth' in the path analysis figure.

Line 245: Cites table S1 but I think it should be Table S2. It would be good to add to Table S2 (and Fig 3) the interpretation and probably the % of variation explained by each of the four foraging PCs, if not it is hard to interpret the differences in the results for each.

Line 262: I am not sure about the current 'body size' correction of including the term in the path analysis but just interacting with beak size. The idea with my previous comment was that beak size could be affected by the effect of climate on overall body size.

Line 377: To clarify the authors could explicitly state what the reverse effect would be, eg. selection for song traits that affect beak morphology? Is there any evidence of this? As suggested previously by Reviewer 3, some references would be good here.

Fig 1. I would probably move A to supplementary material and put as Fig 1 current Fig S2, adding the B part of current Fig 1 to the axes.

Fig 4. It looks good, thanks for adding that, I think it summarises well all the results.

It would be great if all supplementary tables and figures could be in the same file, much easier to go through them and compare.

Review form: Reviewer 3 (Matthew Symonds)

Recommendation

Accept with minor revision (please list in comments)

Scientific importance: Is the manuscript an original and important contribution to its field?

Excellent

General interest: Is the paper of sufficient general interest?

Good

Quality of the paper: Is the overall quality of the paper suitable?

Excellent

Is the length of the paper justified?

Yes

Should the paper be seen by a specialist statistical reviewer?

No

Do you have any concerns about statistical analyses in this paper? If so, please specify them explicitly in your report.

No

It is a condition of publication that authors make their supporting data, code and materials available - either as supplementary material or hosted in an external repository. Please rate, if applicable, the supporting data on the following criteria.

Is it accessible?

No

Is it clear?

N/A

Is it adequate?

N/A

Do you have any ethical concerns with this paper?

No

Comments to the Author

I was Reviewer 3 from the first time this paper was reviewed. I think the authors have done a very thorough and thoughtful revision. I appreciate the lengths they have gone to to attempt different types of analyses, and compare different models. This all greatly increases confidence in their results and conclusions. They have also addressed the issues surrounding the comparisons of size and shape. I have a few remaining issues, but nothing I think that should cause too much lost sleep. Overall, this is a very impressive analysis, the result of clearly a huge amount of work. It is novel and significant in the sheer depth in which the authors attempt to understand variation in beak shape and size, and will I think be a well-cited study (I will certainly cite it anyway!).

Methods:

Morphometrics – all measures were done by the first author, but given there are quite a few other potential sources of error (e.g. exact position of the birds in pictures) – do you have any information on the repeatability? This isn't essential (since the error would just add noise, I think) – but if you do have this information it would be good to present it (very briefly).

Climate measures – line 175: over what time period are the temperatures averaged?

Line 199 – estimate lambda parameters. I think it would be good to (very briefly) state what the function of lambda is (e.g. something like “it controls for the amount of phylogenetic effect in the model residuals”).

Line 199-200 – The variables were ‘scaled’ – was this simply by dividing by the sd (so that all variables had the same sd = 1)? or did you also centre/z-standardise (i.e. subtract the mean and then divide by the sd?).

Lines 270-272: I appreciate that the authors have gone to some effort to also consider non-linear relationships – but given that in one case they DO find that a non-linear fit is better, it would be helpful to explain what the nature of the relationship is (i.e. describe the non-linearity – maybe even visualise with a figure in supplementary material?).

In Figure S4 – contrary to the caption most of the correlation coefficients are below the arrows, not above them

Figure S7 caption – fo should be for

Lines 342-347: I like the recognition that other factors might mitigate the effects of diet or temperature on beak shape and size. Another one that could be considered here is that there is likely variation in physiological control of heat loss (i.e. the capacity to vasodilate blood vessels in the beak), independent of beak morphology – see for example Tattersall et al.'s (2018) on Darwin's finch species - *Functional Ecology* 32: 358-368

Decision letter (RSPB-2019-2474.R0)

15-Nov-2019

Dear Dr Friedman

I am pleased to inform you that your manuscript RSPB-2019-2474 entitled "Evolution of a multifunctional trait: shared effects of foraging ecology and thermoregulation on beak morphology, with consequences for song evolution" has been accepted for publication in Proceedings B.

The referees have recommended publication, but also suggest some minor revisions to your manuscript. Therefore, I invite you to respond to the referees' comments and revise your manuscript. Because the schedule for publication is very tight, it is a condition of publication that you submit the revised version of your manuscript within 7 days. If you do not think you will be able to meet this date please let us know.

Sincerely,

Dr Sasha Dall
mailto: proceedingsb@royalsociety.org

Associate Editor
Board Member
Comments to Author:

I appreciated the thoughtful and thorough revision and response to the reviewer comments, as did the two reviewers who had both previously reviewed the manuscript. After reading the revised manuscript in detail, the reviewers have a number of suggestions, all of which are minor and can be easily addressed. I have nothing further to add to these comments, except that I was not able to access the Dryad data file of the traits - (no DOI provided). It is a condition of publication that authors make their supporting data, code and materials available, so please ensure that you do. This study makes a very nice contribution to our understanding of beak shape variation in birds, and more generally, the evolutionary drivers of morphological traits with multiple functions.

Reviewer(s)' Comments to Author:

Referee: 1

Comments to the Author(s).

I appreciate the effort that the authors have made to follow our suggestions in the previous round of reviews. The analyses are explained much more clearly and I am satisfied with most of the responses to my comments and suggestions.

After reading again the manuscript I have a few more comments to add:

Line 87: I am not sure if this whole 'speciation' framework is needed, or adds much... the current study is far from informing on the divergence aspect of things. I think is already interesting enough with just saying that it can affect species interactions by affecting song traits. Just a thought...

Line 90: But they are oscines as well, the sentence sounds confusing, maybe reword?

Line 101: 'Song behaviour' feels strange, maybe song traits or characteristics?

Line 150: 'Naturalist'?

Line 204: I was confused by the number and thought there was a mistake. Maybe better to not to begin the paragraph with the number.

Line 212: In the text and revisions the authors mention they ended up using residuals too but I couldn't find the results from that analysis. Given that reviewer 3 had concerns regarding the use of residuals maybe best to leave out? Please update properly.

Line 227: Please explain what the PLS analysis will be used for, what will it test?

Line 243: Is this PC1 of elongation the same PC of 'Depth' in the path analysis figure? If so please name the same way (and add in the figure that it is a PC). I still find confusing why they are called foraging PCs or shape PCs in some tables but just 'Nectar', 'Depth' in the path analysis figure.

Line 245: Cites table S1 but I think it should be Table S2. It would be good to add to Table S2 (and Fig 3) the interpretation and probably the % of variation explained by each of the four foraging PCs, if not it is hard to interpret the differences in the results for each.

Line 262: I am not sure about the current 'body size' correction of including the term in the path analysis but just interacting with beak size. The idea with my previous comment was that beak size could be affected by the effect of climate on overall body size.

Line 377: To clarify the authors could explicitly state what the reverse effect would be, eg. selection for song traits that affect beak morphology? Is there any evidence of this? As suggested previously by Reviewer 3, some references would be good here.

Fig 1. I would probably move A to supplementary material and put as Fig 1 current Fig S2, adding the B part of current Fig 1 to the axes.

Fig 4. It looks good, thanks for adding that, I think it summarises well all the results.

It would be great if all supplementary tables and figures could be in the same file, much easier to go through them and compare.

Referee: 3

Comments to the Author(s).

I was Reviewer 3 from the first time this paper was reviewed. I think the authors have done a very thorough and thoughtful revision. I appreciate the lengths they have gone to to attempt different types of analyses, and compare different models. This all greatly increases confidence in their results and conclusions. They have also addressed the issues surrounding the comparisons of size and shape. I have a few remaining issues, but nothing I think that should cause too much lost sleep. Overall, this is a very impressive analysis, the result of clearly a huge amount of work. It is novel and significant in the sheer depth in which the authors attempt to understand variation in beak shape and size, and will I think be a well-cited study (I will certainly cite it anyway!).

Methods:

Morphometrics – all measures were done by the first author, but given there are quite a few other potential sources of error (e.g. exact position of the birds in pictures) – do you have any information on the repeatability? This isn't essential (since the error would just add noise, I think) – but if you do have this information it would be good to present it (very briefly).

Climate measures – line 175: over what time period are the temperatures averaged?

Line 199 – estimate lambda parameters. I think it would be good to (very briefly) state what the function of lambda is (e.g. something like “it controls for the amount of phylogenetic effect in the model residuals”).

Line 199-200 – The variables were ‘scaled’ – was this simply by dividing by the sd (so that all variables had the same sd = 1)? or did you also centre/z-standardise (i.e. subtract the mean and then divide by the sd?).

Lines 270-272: I appreciate that the authors have gone to some effort to also consider non-linear relationships – but given that in one case they DO find that a non-linear fit is better, it would be helpful to explain what the nature of the relationship is (i.e. describe the non-linearity – maybe even visualise with a figure in supplementary material?).

In Figure S4 – contrary to the caption most of the correlation coefficients are below the arrows, not above them

Figure S7 caption – fo should be for

Lines 342-347: I like the recognition that other factors might mitigate the effects of diet or temperature on beak shape and size. Another one that could be considered here is that there is likely variation in physiological control of heat loss (i.e. the capacity to vasodilate blood vessels in the beak), independent of beak morphology – see for example Tattersall et al.'s (2018) on Darwin's finch species - *Functional Ecology* 32: 358-368

Author's Response to Decision Letter for (RSPB-2019-2474.R0)

See Appendix B.

Decision letter (RSPB-2019-2474.R1)

21-Nov-2019

Dear Dr Friedman

I am pleased to inform you that your manuscript entitled "Evolution of a multifunctional trait: shared effects of foraging ecology and thermoregulation on beak morphology, with consequences for song evolution" has been accepted for publication in Proceedings B.

Open Access

Paper charges

Sincerely,
Proceedings B
mailto: proceedingsb@royalsociety.org

Appendix A

Associate Editor

Board Member: 1

Comments to Author:

This study shows that different selection pressures drive different aspects of beak morphology (specialisation along different functional axes): climate has strongest influence on beak size, foraging ecology has strongest influence on beak shape and beak shape is also correlated with vocal performance (song characteristics). A strength of the study is that it assesses multiple selection pressures/potential functions whereas most studies test a single function for a given morphological feature. All three reviewers agree that the study would be of broad interest to readers of PRSLB. However, all three reviewers raised some serious methodological concerns that would need to be addressed before the paper could be published. In particular, clarification is needed on how beak size was measured, and how body size was controlled for in the path analysis. Reviewer 1 also has some good suggestions for aspects that could be developed further, particularly in relation to beak shape and song. If these methodological issues are addressed the study will make a strong contribution to our understanding of morphological evolution.

We appreciate the editor and reviewers for the time and effort they put into the peer review process. We revised our methods section to clarify how beak size was measured, and how we corrected for body size throughout. Each method of correcting for body size has strengths and weaknesses, and there was not consensus among reviewers which method was preferred, so we performed all of them and showed that the results are qualitatively similar. We have added additional analyses suggested by Reviewer 1 and Reviewer 2, as well as citations and clarifications suggested by Reviewer 3.

Reviewer(s)' Comments to Author:

Referee: 1

Comments to the Author(s)

Overall I think this is a great study that combined extensive datasets from different origins to explore very interesting questions regarding the evolution of beaks in birds. The authors compare the role of different selective pressures in the evolution of beak shape in honeyeaters (Meliphagidae) and find that foraging ecology and climate affect differently two aspects of beak morphology.

I think the paper is written very clearly, and the authors have done a great job in condensing and summarising a great amount of information to make it accessible to the reader. I only have one issue with that, and is that the statistical analyses seem to be too simplified and there could be details missing that some readers would like to have, in case someone wants to do similar analyses. I hope the authors find the following comments useful:

1. It wasn't clear to me what type of model they used for the multivariate analyses, they cite Adams (2014), was *geomorph* used? If so, would be good to specify. Also good to mention explicitly which were the predictor variables used, because there are different sets of predictors in the path analysis described previously.

We appreciate this request for clarification, and after reading the paragraph again we completely agree. We have added text to this paragraph to describe in greater detail the method we used for this analysis, which is implemented in *geomorph* as reviewer 1 expected. We now specify the model (λ), parameters, and predictor variables included in the analysis. Details of the multivariate PGLS analyses can now be found in supplemental table 2.

Added (Lines 251–260): “ We performed a phylogenetic Procrustes ANOVA that treated Procrustes-aligned beak shape as a response variable. ... We performed this analysis using a

Brownian motion model in the *geomorph* function *procD.pgls*, and 5000 iterations of resampling for significance testing.”

2.It wasn't clear to me either how body size was controlled for in the path analyses. The authors mention that in PGLS analyses body and beak size were not included in the same model because of collinearity, I imagine the same logic was applied to the path analyses. However, would this mean that the effects of temperature on beak size reported in the path analyses are actually effects on body size? Would it be possible to use the residuals of beak size (controlled by body size) as variable in this analysis? I guess it becomes a question of whether climate/nectarivory affect relative or absolute beak size.

As is described in the PGLS analysis results (now Table S2), body size was significantly correlated with beak size in the expected allometric relationship, but was not correlated with any of the other beak morphology response variables in this study. Adding body size as an additional variable to an already busy model increased the total number of parameters considerably, which we wanted to avoid. Consequently, for our new submission we opted to follow the reviewer's suggestion of running an additional path analysis using beak size residuals (from their log-log regression against body size) rather than including body size as an additional predictor. This increased the strength of the relationships we observed between beak size and each of its functional predictors. We report these results alongside results from non-residual data, due to a concern about this approach brought up by Reviewer 3.

3.I understand the logic in arguing that multifunctional trait model has a higher fit than single function models (according to the path analysis). Maybe it would be good to clarify that the statistic used for comparison (CICc) takes into account the number of paths, and models with more paths are not always favoured (e.g. in general we would expect a model with more parameters to explain more variation than a model with less parameters (e.g.paths)).

We appreciated this suggestion, and we now include such a clarification in the text (lines 247–249):

“We evaluated these models by comparing their (CICc) statistics, which describe model fit while taking into account the number of parameters/paths [60]”

4.The authors suggest that nectarivory contributes more than climate in driving the evolution of beak shape (curvature and depth). However, the path analysis shows that climate can significantly affect nectarivory, and so climate could have an indirect effect on beak shape through nectarivory. It would be interesting to calculate, compare and discuss these indirect routes, to me this is one of the interesting features of path analyses. Also, in figure 2, are nectar, curvature, depth different PCs (those in Fig 2C)? If so, it would be good to clarify that in text and in the fig (e.g. Nectar PC1).

We added an extra figure to the supplement describing total effects (i.e., including indirect effects; Figure S3). We have also added text to the title for Figure 2 to clarify which PC axes we are referring to.

5.It would be nice to see a stronger connection between the analyses on song and the other analyses. This is mentioned briefly on lines 330-335 but I think the point could be stronger. Maybe it's too much, but as a suggestion, the authors could make a figure combining the findings from all the analyses. Like a simplified path graph showing how climate/foraging variables affect which beak traits and which beak traits affect which song traits. I think this is one of the most interesting points of this study and could be made stronger. I also think that figures 1a and 4a are not totally necessary and could be moved to supplementary material in case the authors want more space.

We added a simplified path diagram summarizing the results of our study, as per the reviewer's suggestion. This is currently shown as a panel in Figure 4, but could be separated if that improves the flow or arrangement of the article when it's laid out.

Likewise we've moved 1c and 4a to the supplement.

Minor comments:

Line 26: I think the 'while' could be replaced with a 'however' after the comma.

We made this correction.

Line 118: How many species?

We added the number of species as well as sampling effort to this sentence.

Figure 1: The picture looks like it is on an angle and it is not a standard one (e.g. 90), which made me wonder how the authors could consistently take pictures from that same angle (e.g. slightly from the side). Is this an actual picture used? If so, maybe clarify where pictures were taken from and whether the angle was consistent.

Because I couldn't think of another way, I'm going to describe the angles of rotation for the bird as though it were flying like an airplane: "roll" dipping the wing to the left or right, "pitch" rotating the bird's nose up or down, and "yaw" rotating as a whole to the left or right.

Standardized photos were taken in profile (Figure 1). As we now describe in a little more detail, an adjustable stage was used to prevent rotation out of the proper 2D plane either as "yaw" or as "roll" (which would have been a problem).

The generalized Procrustes alignment corrects for differences between specimens in "pitch", the rotation of the shape in its proper 2D plane, by minimizing the distance between landmark of the specimens. So the rotation in the picture is no problem for our analysis, and we've adjusted Figure 1 so it's not a distraction.

Line 217: It would be good to present a table in the supplementary material with the specific results of the PGLS models (e.g. Beta, T-value, P value).

We now include such a table in the supplement, as Table S2.

Line 255: I don't understand how it suggests that selection on these functions is evident during diversification, please clarify.

We have rephrased this as "over evolutionary timescales".

Overall it was a very nicely written and interesting paper, congratulations.

Thank you very much, we've been working on this a long time, so it's very nice to read this.

Referee: 2

Comments to the Author(s)

I have reviewed the manuscript entitled "Evolution of a multifunctional trait: shared effects of foraging ecology and thermoregulation on beak morphology, with consequences for song evolution" and I find it to be a sound and interesting paper. The premise of examining many predictors of beak shape is rare, as the authors present in the thorough literature appraisal of the introduction. The paper is well written and enjoyable to read. The data are impressive and sufficient to answer the question, and the analytics are mostly appropriate – with one issue detailed below that will possibly change the

outcomes.

My main concern regards the treatment of the foraging behaviour data. A ‘phylogenetic’ PCA is not appropriate here because this serves to “double-correct” the data when coupled with a PGLS. You have not done a phylogenetic-PCA for the shape data, so why do so for the diet data? Phylogenetic PCAs are often mistreated in analyses; since PCA is not a statistical approach that assumes independence of the input variables, it does not need to be explicitly altered to account for the statistical non-independence of species data. A standard PCA would suffice for reducing dimensionality of the diet data if it was strictly necessary to do so.

We have to respectfully disagree on this point. The original paper for Liam Revell’s phylogenetic PCA (Revell 2009) recommends it for use prior to subsequent analysis with PGLS. As you may have noticed, PCA can perform poorly (creating for example a horseshoe effect with one biologically meaningless axis) when species have non-linear relationships (i.e., a phylogeny). The phylogenetic PCA corrects for these kinds of issues, but importantly it does not correct for phylogenetic non-independence, which is a separate issue. So we are not double-correcting. While it would be possible for us to do an ordinary PCA to go along with the reviewer’s suggestion, a previous study that our work builds upon (Miller et al. 2017) uses a phylogenetic PCA, and keeping our methods congruent will greatly improve readers’ ability to interpret the results of our paper in light of previous research.

Since foraging behaviour is a matrix of variables considered together, as is the Procrustes residuals of shape data, they are suitable for multivariate methods. As a complement to the path analysis, I suggest the authors consider the two-block partial least squares analysis (2B-PLS) to consider the patterns of covariation between the different factors and beak shape. For example, one can examine beak shape (one ‘block’, aka matrix, of the Procrustes residuals) and foraging behaviour data (second block, the diet matrix), and show which behavioural traits influence specific shape changes. This is a standard technique for looking for axes of covariation between two multivariate datasets (Rohlf FJ & Corti M, 2000. The use of partial least-squares to study covariation in shape. *Systematic Biology*), and this can be done in a phylogenetic context (Adams, D.C. & Felice, R.N., 2014. Assessing Trait Covariation and Morphological Integration on Phylogenies Using Evolutionary Covariance Matrices. *PLoS ONE*) and implementable in the same R package used, geomorph. The strength of covariation can then be compared to covariation of shape with other multidimensional blocks (song characteristics, climate) through multiple 2B-PLS analyses, and using an approach that is presented as a way to compare patterns of morphological integration (Adams, D.C. & Collyer, M.L., 2016. On the comparison of the strength of morphological integration across morphometric datasets. *Evolution*), but that simply serves as a method to compare covariation coefficients across different comparisons in any type of data.

We ran a two-block PLS analysis comparing shape and foraging behavior, which supported the strong correlation between these traits. These results are now described in lines 326–328.

I see that the authors mention “multidimensional partial least squares analysis” in the caption of Figure 3 (line 657), but with the wrong citation (55, that for multidimensional PGLS). But with no mention of the PLS in the methods, I am not at sure what they have done. This needs to be clarified also.

We revised this to clarify that this was indeed a multidimensional PGLS, which we now refer to as a Procrustes PGLS throughout the manuscript.

The authors seem to swap between using the whole shape data matrix, and the PC axes on their own in statistical analysis, which is confusing to follow. Please change the sentence Line 127-128, because they are not always being used in subsequent analyses, and be explicit where these PC axes are being used as separate variables, and where the Procrustes residuals shape matrix is being used.

We agree that this creates confusion, and we re-wrote much of the comparative methods section to try and address this issue. We appreciate any and all help in identifying where it's unclear which method is used.

Only the upper part of the beak was measured in this study. I assume the lower part is not standard in shape or size, and therefore the study is missing an important aspect of the overall beak shape. A comment on how this may affect the results and interpretations would be good to see.

This was an unfortunate constraint of the study skin material we used. We describe the limitations of this in the methods (lines 129–133), and discuss how this issue can be addressed in the future.

Line 127 – Principal not principle

We fixed this mistake.

Figure 1 – PC axes must have values on the tick marks. Would also be good to have % of variance on each PC axis label.

The figure has been updated accordingly.

Referee: 3

Comments to the Author(s)

In this paper, as the title indicates, the authors demonstrate how foraging ecology and temperature influence elements of beak morphology using data from Australasian honeyeaters. On the plus side, it's really nice to see a paper consider BOTH diet/foraging strategy and thermoregulation and their relative influence on beak morphology. This is good because, in my experience, many ornithologists consider the role of temperature in driving changes in beak size to be miniscule in comparison with foraging ecology. Therefore, it's important to see a paper demonstrate this not to be the case. I find the authors main conclusions basically convincing. The addition of the analysis investigating the effects of beak morphology on song is interesting.

However, the paper at present does have quite a number of issues in regard to the methods and results/interpretation which need to be addressed, which I found dampened my enthusiasm for the study. I think (or hope) that these concerns can be addressed through some more careful explanation/wording, but because of the lack of clarity I can't entirely rule out that the analyses and interpretation have some more fundamental flaws.

Major concerns:

The paper makes clear how elements of beak shape are estimated (through PCs), but is very unclear as to how beak size is actually calculated, despite this being a major part of the paper. The only mention of beak size in the methods is in the final few lines, in relation to analysing its effect on song structure, but with no detail of how it is measured. Also since the beak measurements appear to only consider a proportion of the upper beak, I wonder to what extent beak size has been accurately estimated (although without further detail it's impossible to know if this is a fair criticism). On a more minor level, this issue of how beak size is actually defined also affects the introduction in lines 71-73, where individual elements beak depth and beak length, are described as if they are independent of beak size (I'd suggest simply describing these measures as 'elements of beak shape', which they also are).

We now clarify in our methods that we used the centroid size of our Procrustes-aligned beak landmarks to estimate beak size. As the reviewer has pointed out, we were only able to use a

portion of the beak for these landmarks – from the apex to the nare – since feathers obscure the proximal edge of the beak in photographs of many species. This problem, like the mandibular occlusion mentioned above, may be ameliorated in the future by use of penetrative scanning or skeletal material. In the museum, we also took caliper measurements of beak length, width, and depth. The first principal component of these measurements, which is typically and in this case beak size, was highly correlated with beak centroid size ($p < 0.001$, $r^2 = 0.87$).

We also liked the suggestion to replace “axes of variation” with “elements of beak shape”, and made this change throughout the manuscript.

The results present some difficulties in interpretation, in part because two different approaches are used – phylogenetic path analysis, and then, in order to generate pseudo-R² values, a multivariate phylogenetic regression (by which I assume it to mean that ‘shape’ generally is the response variable, rather than the individual components of shape (elongation and curvature – though the former is confusingly called ‘Depth’ in Figure 2e). One problem here is that the effects on beak size are not clearly spelt out – merely that climate is a stronger predictor (as judged from Figure 3a) than foraging ecology. There is one mention (line 308) that the size of beaks increases with warmer temperatures. However, Figure 2e presents a confusing picture in that winter temperature is positively associated with beak size (i.e. warmer climates = bigger beaks), but summer temperatures are negatively associated (i.e. warmer climates = smaller beaks). I appreciate that in part this has been more explicitly investigated in the authors’ previous paper in *Evolution* – where winter temperature is a better predictor than summer temperature. However, the confusing thing here is that Figure 2e indicates that the effects of winter and summer temperatures are both significant, but Fig 3a indicates that only winter temperature has a significant effect. However, in this case why is there a discrepancy between the two analyses. I think the problem here may lie in not having explained how the measure of beak size is generated – but in any event more discussion is needed in the text of the way in which climate actually influences beak size (see a bit more discussion below). There is also some confusion when comparing the results of Figure 2e and Figure 3a, in linking the foraging ecology PCs with ‘Nectar’. To me it seems like the model on which the analysis of Figure 3a is based is different from the model in figure 2e – in which case shouldn’t more details of the model in 3a be given, rather than just R² values?

We hope that clarifying in greater detail how beak size was measured helps to resolve some of this concern, in conjunction with our reorganization of the methods and addition of a total effect plot from the path analysis in the supplement. What seems to be going on here is that high summer temperatures are highly correlated with nectarivory. This is likely to be inflating the effect of summer high temperatures on beak shape, which are low when this relationship is corrected for (Fig. S2). As suggested by this and another reviewer, we added a sentence to point readers to these indirect effects, and revised our section describing these results (lines 344–346; see below). Hopefully this helps to clear up the discrepancies between Figs. 2 and 3.

“However, when including and correcting for indirect effects (summer temperatures predicting nectarivory), winter temperatures showed a greater effect on beak morphology (Figure S4)”

On a related note at lines 234–235 it is stated that the effect of summer temperature on shape is greater than that of winter temperature – presumably based on the results in Fig 3b, however, this does not appear to be supported by Fig 2e, where the correlation coefficients for summer temperature with shape elements is smaller in magnitude than that link winter temperature to depth. I think I know what the answer to this is (you are referring in 3b to shape overall, rather than elements of shape) but I still think this should be explicitly discussed.

We now clarify that this is referring to the results from our Procrustes PGLS analysis on overall shape, and following a suggestion from another reviewer add text noting that this was contradicted by the total effect in our path analyses (Fig. S2).

Regarding the contradiction between the effects of summer temperature and winter temperature, might the negative (or lack of significant relationship) with the former be explained by very hot environments causing selection for SMALLER beaks due to them becoming a liability in taking on heat when ambient temperatures elevate beyond the bird body temperature. Greenberg & Danner (2012) Evolution demonstrated how curvilinear relationships of beak size with summer temperature can result from this. It would be interesting to either examine whether there is evidence for any curvilinear relationship, or at least get some idea of the range of maximum summer temperatures experienced by the species in the analysis.

Following the reviewer's suggestion, we have added polynomial regressions to the supplementary material. In most cases, and especially with beak size, curvilinear relationships with temperature were supported as a better fit to our data than linear relationships. We also added a sentence reporting the ranges of average winter minimum and summer maximum temperatures across species in this study (lines 346–347).

Line 122 – two questions raised here: the measurement area of the beak is not entirely clear – did you just examine the area of the upper beak anterior to the nares? And also how did you align the angle of the beak towards the camera so that this was consistent - this would surely affect the judgement on the landmarks?

1) You are correct that we only measured the portion of the beak anterior to the nares. We photographed and landmarked the rostrum in its entirety. However, our preliminary analysis showed a great deal of noise in the landmarks to the posterior portion of the beak, caused by difficulty in accurately landmarking the posterior apex of the rostrum. We have added text to our methods section to make this clear, and offer a few suggestions as to how future studies may overcome this issue.

2) We now describe the use of an adjustable stage for this purpose in our methods section. To add some more detail to that, we set an adjustable stage with a measurement standard attached to match the midsagittal plane of the specimen (to control for “yaw” of the specimen), and rotated the specimen along its central axis until the midsagittal plane of the beak was in line with the stage and standard (to control for the “roll” of the specimen). Once the specimen was in line, it would also be in line with the camera's focal plane.

Line 180 – how did you deal with intraspecific variation (or at least species sample size) in the analysis? This is something that should be taken into consideration (see chapter by Garamszegi in the book on Modern Phylogenetic Comparative Methods).

We now discuss this in the second and third paragraphs of our methods section, and report our average sample size. We dealt with intraspecific variation by using species averages. While phylogenetic comparative methods that correct for intraspecific variation are to be preferred as you describe, to our knowledge they are not yet implemented for the phylogenetic path analysis, Procrustes PGLS, or the partial least squares analysis that we performed. However, since singing behavior is widely acknowledged to be variable within species, especially among passerines, we repeated our PGLS analysis comparing song and beak morphology using a method that accounted for intraspecific variation in song traits.

I was a bit concerned about how the authors dealt with body size as a controlling factor in the analysis of song evolution. Because of the high correlation between beak size and body size they did not include both simultaneously in any of the PGLS analyses. For one thing, it's not clear how it was decided 'a priori' (line 200) which was the more relevant.

We repeated our analyses using beak size residuals from their allometric regression on body size, and report these similar results.

We also clarified that this was based on previous research, citing Podos 2001, Nowicki & Westneat 1992, and now also Ryan and Brenowitz 1985, in support of predicted relationships between beak size and pace, and body size and minimum frequency. We believe the logic behind these assertions has been established previously. Larger animals have larger vibrating internal structures, which produce vibrations of longer wavelength and thus lower frequency (Beranek 1954). Likewise, a larger extremity requires more work to move back and forth rapidly, as sparrows rapidly adjust their beak gape when changing pitch during a trill (Nowicki & Westneat 1992).

Additionally, the issue here is that it's not therefore possible to evaluate to what extent it is body size or beak size that is driving song evolution. Because using residuals values is not appropriate (see Freckleton 2002 *J Anim Ecol*), it might be suggested to still include both variable in the model as suggested by Freckleton. If the authors don't feel this is valid (I must admit I've not seen this issue addressed specifically in response to Freckleton's paper), then at least I'd suggest comparing models including either body size or beak size and see which provides a better fit for the data.

In both of these cases, we are trying to estimate the effect of size on song behavior. We are also trying to correct for it, so as to tease out the effect of beak shape on song behavior. We've got to correct for it somehow, but correcting for it using both beak size and body size at once creates an unreliable regression model for our purposes (estimating effects of individual predictors) because these size variables have near-perfect collinearity. However, as described above, one of these variables is much more relevant in the proximate mechanisms of producing different frequencies, and one of these variables is much more relevant in the proximate mechanisms of producing songs at different paces. Note also that Grant and Grant (2002) used a similar statistical approach to this beak/body size collinearity issue.

To satisfy the reviewer's concerns while avoiding the collinearity issue, we repeated our analysis correcting for body size and beak size in different regression models (see Fig. S3). In the body-size corrected model, beak shape PC1 (depth/elongation) is correlated with pace.

We are aware of this issue of using body size residuals, and our methods were organized in part to avoid it in our first submission. However, after a recommendation from another reviewer, we have repeated our analyses using body-size residuals in cases when it was not possible to correct for allometric effects in a more appropriate way. Please note that our results are qualitatively the same either way (no changes in significance level).

Line 154-156 – this last sentence is not clear (the e.g. in brackets does not explain what you are getting at here).

We revised this sentence to be more specific:

“We included intervals greater than 1 second in songs only if they were part of a consistently repeated pattern of note types (e.g., A ... BC).”

To clarify for the reviewer, we wanted our operational definition of a song to be inclusive of song behaviors we observed in honeyeaters. Some recordings included intervals that were greater than 1 second, but were obviously part of a song structure repeated throughout the recording. In the example, if we consider A, B, and C to be unique note types (this ABC notation being fairly standard in vocal behavior literature), a species would repeat all three note types with a second or two gap between A and B. Listening to a recording like this we would write down A...BC A...BC A...BC A...BC) and call it a single repeated song, as opposed to A and BC being different alternating songs. We're hoping to go into more detail about honeyeaters' song structure in future studies.

Can you be certain that beak shape/size drives song evolution and it's not the other way round (i.e.

selection on song is driving changes in beak shape/size) – some reference here as to why it's definitely the other way round would be good.

This is a good point, and we added a caveat to this effect in the discussion (lines 520–521)

Lines 211-213 – this analysis of phylogenetic signal is not covered in the methods, nor is the reason for it justified/explained.

We removed this part about phylogenetic signal, since we're not really doing much with it.

Minor things:

Line 27: a very minor thing – but I would write this as more simply as Bird beaks, rather than Birds' beaks

We've made this change.

Line 59: when referring to preening you could also mention that beak shape is related to efficient parasite removal – see Villa et al. (2018) *Evolutionary Ecology* 32: 443-452

We've added text to include parasite removal as suggested, but retain the original citation focused on parasite removal.

Line 60: there is a more recent extensive review of the role of the avian beak in thermoregulation which would be appropriate to cite here: Tattersall et al. (2017) *Biological Reviews* 92: 1630-1656

We've replaced this citation as recommended.

Lines 76-78: You should also probably discuss somewhere the new paper from the same group, which finds complex relationships between beak shape and feeding ecology - Navalon et al. (2019) *Evolution* 73: 422-435

We've added this citation as recommended.

Line 98: I don't know what you mean by "a conveniently controlled replicate of the evolutionary experiment". I'd drop this expression.

As this group has evolved multiple times into environmental, morphological, and ecological spaces all within the same biogeographic region, it gives us an opportunity to evaluate evolutionary linkages between climate, diet, morphology, song, and beak shape in the same spatial arena. We've tweaked this sentence to make this more clear.

Line 101: It may be worth pointing out somewhere that Gardner et al (2016) *Climate Change Responses* 3: 11 found no direct link between temperature and bill size in Meliphagoidea, but did find that temperature mediates the effect of humidity on bill size - in general it would be worth including some (brief) discussion later of the effect of humidity on bill size.

We've added this citation as recommended.

Methods section a) this seems curiously placed here right at the start of the methods. I would include the information about phylogeny in the later section on comparative methods where it is more directly relevant.

We agree and have changed the order accordingly.

Line 181 – it would be to explain what lambda is for those readers who do not know.

If we were using the lambda parameter for something like an estimate of phylogenetic signal, we would happily add a sentence describing its purpose. However, since it's only used here as an intermediate step in the workflow and reported for repeatability's sake.

Lines 257-260 - this seems an odd comparison/statement. implies that comparing to the Galapagos finches was intended as a key aim of the study in the first place. The statement seems a bit redundant (even if is correct).

This is a valid criticism, so we removed the paragraph entirely.

Line 438 – reference 17 doesn't seem like it is correctly given (missing volume number and real page numbers?)

We have corrected this mistake.

Line 518 – there is a little 'a' after Tobias J – presumably this should be Tobias JA.

We have corrected this mistake.

Line 664 – Standardised beta values - Does this mean the variables were standardized prior to analysis, or that you standardised after calculation of the beta values (if so, how? This information should be given in the methods).

We now clarify in the methods (lines 274–275) that variables were scaled to produce comparable effect sizes.

Appendix B

Associate Editor

Board Member

Comments to Author:

I appreciated the thoughtful and thorough revision and response to the reviewer comments, as did the two reviewers who had both previously reviewed the manuscript. After reading the revised manuscript in detail, the reviewers have a number of suggestions, all of which are minor and can be easily addressed. I have nothing further to add to these comments, except that I was not able to access the Dryad data file of the traits - (no DOI provided). It is a condition of publication that authors make their supporting data, code and materials available, so please ensure that you do. This study makes a very nice contribution to our understanding of beak shape variation in birds, and more generally, the evolutionary drivers of morphological traits with multiple functions.

We really appreciate the editor and reviewers' time and effort in conducting a thorough review of this manuscript.

Our analysis and data should now be properly linked to the manuscript.

Reviewer(s)' Comments to Author:

Referee: 1

Comments to the Author(s).

I appreciate the effort that the authors have made to follow our suggestions in the previous round of reviews. The analyses are explained much more clearly and I am satisfied with most of the responses to my comments and suggestions.

After reading again the manuscript I have a few more comments to add:

Line 87: I am not sure if this whole 'speciation' framework is needed, or adds much... the current study is far from informing on the divergence aspect of things. I think is already interesting enough with just saying that it can affect species interactions by affecting song traits. Just a thought...

We are hoping this will highlight one of the implications of our findings, which we expect will be of interest to researchers working on divergence along ecological and reproductive lines.

Line 90: But they are oscines as well, the sentence sounds confusing, maybe reword?

We made a few tweaks to this sentence, but we are still are trying hard here to avoid describing the Meliphagidae as “basal” oscines.

Line 101: 'Song behaviour' feels strange, maybe song traits or characteristics?

We have made this change following the reviewer's recommendation.

Line 150: 'NaturalisT'?

It really is Naturalis, we promise!

Line 204: I was confused by the number and thought there was a mistake. Maybe better to not to begin the paragraph with the number.

We moved these numbers so they no longer lead their paragraphs.

Line 212: In the text and revisions the authors mention they ended up using residuals too but I couldn't find the results from that analysis. Given that reviewer 3 had concerns regarding the use of residuals maybe best to leave out? Please update properly.

Since both methods have strengths and weaknesses, we opted to present the results from both in Figure 2e; values for body-size residuals are given above arrows, and values for non-corrected results are given below. An alternative model is presented in Figure S4.

Line 227: Please explain what the PLS analysis will be used for, what will it test?

We added a sentence clarifying this point: "This analysis was intended to test for and measure the degree of overall covariation between these highly dimensional traits."

Line 243: Is this PC1 of elongation the same PC of 'Depth' in the path analysis figure? If so please name the same way (and add in the figure that it is a PC). I still find confusing why they are called foraging PCs or shape PCs in some tables but just 'Nectar', 'Depth' in the path analysis figure.

This was an error on the authors' part - elongation was changed to depth midway through drafting the manuscript to align the axis polarity with the description. We have edited this sentence to clarify that we are referring to depth here.

Line 245: Cites table S1 but I think it should be Table S2. It would be good to add to Table S2 (and Fig 3) the interpretation and probably the % of variation explained by each of the four foraging PCs, if not it is hard to interpret the differences in the results for each.

We made the recommended changes to the text, and added the % variation explained to Table S2 (in Figure 3 it's confusing because the figure is describing the beak shape variance explained by those variables).

Line 262: I am not sure about the current 'body size' correction of including the term in the path analysis but just interacting with beak size. The idea with my previous comment was that beak size could be affected by the effect of climate on overall body size.

We believe this issue was dealt with exhaustively in the last revision, as we run the path analysis with three sets of terms and found similar results.

Line 377: To clarify the authors could explicitly state what the reverse effect would be, eg. selection for song traits that affect beak morphology? Is there any evidence of this? As suggested previously by Reviewer 3, some references would be good here.

We tweaked the writing here to be more explicit as suggested, but have been having difficulty finding an example of this reverse effect, so it is more a logical than an empirical alternative.

Fig 1. I would probably move A to supplementary material and put as Fig 1 current Fig S2, adding the B part of current Fig 1 to the axes.

Having looked at thousands of beaks during the course of this study I'm a little sad to see the last one leave the paper. The paper is about honeyeater beaks, so I think it's worth having at least one in there. We have a version without the bird, if the editor disagrees.

Fig 4. It looks good, thanks for adding that, I think it summarises well all the results.

This was a great idea and I'll have to remember to include these more often in the future.

It would be great if all supplementary tables and figures could be in the same file, much easier to go through them and compare.

We now organize our supplementary figures and tables accordingly.

Referee: 3

Comments to the Author(s).

I was Reviewer 3 from the first time this paper was reviewed. I think the authors have done a very thorough and thoughtful revision. I appreciate the lengths they have gone to to attempt different types of analyses, and compare different models. This all greatly increases confidence in their results and conclusions. They have also addressed the issues surrounding the comparisons of size and shape. I have a few remaining issues, but nothing I think that should cause too much lost sleep. Overall, this is a very impressive analysis, the result of clearly a huge amount of work. It is novel and significant in the sheer depth in which the authors attempt to understand variation in beak shape and size, and will I think be a well-cited study (I will certainly cite it anyway!).

Methods:

Morphometrics – all measures were done by the first author, but given there are quite a few other potential sources of error (e.g. exact position of the birds in pictures) – do you have any information on the repeatability? This isn't essential (since the error would just add noise, I think) – but if you do have this information it would be good to present it (very briefly).

We don't have this, but we will be sure to put it in the next one.

Climate measures – line 175: over what time period are the temperatures averaged?

We added that this was between 1960 and 1990.

Line 199 – estimate lambda parameters. I think it would be good to (very briefly) state what the function of lambda is (e.g. something like “it controls for the amount of phylogenetic effect in the model residuals”).

We added reviewer's suggested text.

Line 199-200 – The variables were ‘scaled’ – was this simply by dividing by the sd (so that all variables had the same sd = 1)? or did you also centre/z-standardise (i.e. subtract the mean and then divide by the sd?).

We now clarify that we also centered the variables.

Lines 270-272: I appreciate that the authors have gone to some effort to also consider non-linear relationships – but given that in one case they DO find that a non-linear fit is better, it would be helpful to explain what the nature of the relationship is (i.e. describe the non-linearity – maybe even visualise with a figure in supplementary material?).

We've added another figure to the supplementary material to visualize these patterns. It's now Figure S5 and is referred to in the lines the reviewer is referencing.

In Figure S4 – contrary to the caption most of the correlation coefficients are below the arrows, not above them

This was a mistake, thank you for pointing it out.

Figure S7 caption – fo should be for

Again we thank the reviewer for finding this error, which we have now fixed.

Lines 342-347: I like the recognition that other factors might mitigate the effects of diet or temperature on beak shape and size. Another one that could be considered here is that there is likely variation in physiological control of heat loss (i.e. the capacity to vasodilate blood vessels in the beak), independent of beak morphology – see for example Tattersall et al.'s (2018) on Darwin's finch species - *Functional Ecology* 32: 358-368

We have added this citation.